# Novel Class Discovery for Long-tailed Recognition

**Chuyu Zhang**[*]                                          *zhangchy2@shanghaitech.edu.cn*
*ShanghaiTech University, Shanghai, China*
*Lingang Laboratory, Shanghai, China*

**Ruijie Xu**[*]                                            *xurj2022@shanghaitech.edu.cn*
*ShanghaiTech University, Shanghai, China*

**Xuming He**                                               *hexm@shanghaitech.edu.cn*
*ShanghaiTech University, Shanghai, China*
*Shanghai Engineering Research Center of Intelligent Vision and Imaging, Shanghai, China*

**Reviewed on OpenReview:** *https://openreview.net/forum?id=ey5b7kODvK*

## Abstract

While the novel class discovery has recently made great progress, existing methods typically focus on improving algorithms on class-balanced benchmarks. However, in real-world recognition tasks, the class distributions of their corresponding datasets are often imbalanced, which leads to serious performance degeneration of those methods. In this paper, we consider a more realistic setting for novel class discovery where the distributions of novel and known classes are long-tailed. One main challenge of this new problem is to discover imbalanced novel classes with the help of long-tailed known classes. To tackle this problem, we propose an adaptive self-labeling strategy based on an equiangular prototype representation of classes. Our method infers high-quality pseudo-labels for the novel classes by solving a relaxed optimal transport problem and effectively mitigates the class biases in learning the known and novel classes. We perform extensive experiments on CIFAR100, ImageNet100, Herbarium19 and large-scale iNaturalist18 datasets, and the results demonstrate the superiority of our method. Our code is available at `https://github.com/kleinzcy/NCDLR`.

## 1 Introduction

Novel Class Discovery (NCD) has attracted increasing attention in recent years (Han et al., 2021; Fini et al., 2021; Vaze et al., 2022), which aims to learn novel classes from unlabeled data with the help of known classes. Despite the existing methods have achieved significant progress, they typically assume the class distribution is balanced, focusing on improving performance on datasets with largely equal-sized classes. This setting, however, is less practical for realistic recognition tasks, where the class distributions are often long-tailed (Zhang et al., 2021b). To address this limitation, we advocate a more realistic NCD setting in this work, in which both known and novel-class data are long-tailed. Such a NCD problem setting is important, as it bridges the gap between the typical novel class discovery problem and the real-world applications, and particularly challenging, as it is often difficult to learn long-tailed known classes, let alone discovering imbalanced novel classes jointly.

Most existing NCD methods have difficulty in coping with the imbalanced class scenario due to their restrictive assumptions. In particular, the pairwise learning strategy (Han et al., 2021; Zhong et al., 2021b) often learns a poor representation for the tail classes due to insufficient positive pairs from tail classes. The more recent self-labeling methods (Asano et al., 2020; Fini et al., 2021) typically assume that the unknown class sizes are evenly distributed, resulting in misclassifying the majority class samples into the minority ones. An alternative strategy is to combine the typical novel class discovery method with the supervised

---

[*]Both authors contributed equally.

long-tailed learning method (Zhang et al., 2021b; Menon et al., 2020; Kang et al., 2020; Zhang et al., 2021a). They usually require estimating the novel-class distribution for post-processing and/or retraining the classifier. However, as our preliminary study shows (c.f. Tab.2), such a two-stage method often leads to inferior performance due to the noisy estimation of the distribution.

To address the aforementioned limitations, we propose a novel adaptive self-labeling learning framework to tackle novel class discovery for long-tailed recognition. Our main idea is to generate high-quality pseudo-labels for unseen classes, which enables us to alleviate biases in model learning under severe class imbalance. To this end, we first develop a new formulation for pseudo-label generation process based on a relaxed optimal transport problem, which assigns pseudo labels to the novel-class data in an adaptive manner and partially reduces the class bias by implicitly rebalancing the classes. Moreover, leveraging our adaptive self-labeling strategy, we extend the equiangular prototype-based classifier learning (Yang et al., 2022b) to the imbalanced novel class clustering problem, which facilitates unbiased representation learning of known and novel classes in a unified manner.

Specifically, we instantiate our framework with a deep neural network consisting of an encoder with unsupervised pre-training and an equiangular prototype-based classifier head. Given a set of known-class and novel-class input data, we first encode those input data into a unified embedding space and then introduce two losses for known and novel classes, respectively. For the known classes, we minimize the distance between each data embedding and the equiangular prototype of its corresponding class, which help reduce the supervision bias caused by the imbalanced labels. For the novel classes without labels, we develop a new adaptive self-labeling loss, which formulates the class discovery as a relaxed Optimal Transport problem. To perform network learning, we design an efficient bi-level optimization algorithm that jointly optimizes the two losses of the known and novel classes. In particular, we introduce an efficient iterative learning procedure that alternates between generating soft pseudo-labels for the novel-class data and performing class representation learning. In such a strategy, the learning bias of novel classes can be significantly reduced by our equiangular prototype design and soft adaptive self-labeling mechanism. Lastly, we also propose a novel strategy for estimating the number of novel classes under the imbalance scenario, which makes our method applicable to real-world scenarios with unknown numbers of novel classes.

We conduct extensive experiments on two constructed long-tailed datasets, CIFAR100 and Imagenet100, as well as two challenging natural long-tailed datasets, Herbarium19 and iNaturalist18. The results show that our method achieves superior performance on the novel classes, especially on the natural datasets. In summary, the main contribution of our work is four-folds:

- We present a more realistic novel class discovery setting, where the class distributions of known and novel categories are long-tailed.

- We introduce a novel adaptive self-labeling learning framework that generates pseudo labels of novel class in an adaptive manner and extends the equiangular prototype-based classifier to address the challenge in imbalanced novel-class clustering.

- We formulate imbalanced novel class discovery as a relaxed optimal transport problem and develop a bi-level optimization strategy for efficient class learning.

- Extensive experiments on several benchmarks with different imbalance settings show that our method achieves the state-of-the-art performance on the imbalanced novel class discovery.

## 2 Related Work

### 2.1 Novel Class Discovery

Novel Class Discovery (NCD) aims to automatically learn novel classes in the open-world setting with the knowledge of known classes. A typical assumption of NCD is that certain relation (e.g., semantic) exists between novel and known classes so that the knowledge from known classes enables better learning of novel ones. The deep learning-based NCD problem was introduced in (Han et al., 2019), and the subsequent works

can be grouped into two main categories based on their learning objective for discovering novel classes. One category of methods (Han et al., 2021; Zhong et al., 2021a;b; Hsu et al., 2018a;b) assume neighboring data samples in representation space belong to the same semantic category with high probability. Based on this assumption, they learn a representation by minimizing the distances between adjacent data and maximizing non-adjacent ones, which is then used to group unlabeled data into novel classes. The other category of methods (Fini et al., 2021; Yang et al., 2022a; Gu et al., 2023) adopt a self-labeling strategy in novel class learning. Under an equal-size assumption on novel classes, they utilize an optimal transport-based self-labeling (Asano et al., 2020) strategy to assign cluster labels to novel-class samples, and then self-train their networks with the generated pseudo labels.

Despite the promising progress achieved by those methods, they typically adopt an assumption that the novel classes are largely uniformly distributed. This is rather restrictive for real-world recognition tasks as the class distributions are often long-tailed in realistic scenarios. However, both categories of methods have difficulty in adapting to the setting with imbalanced classes. In particular, for the first category of methods, the representation of tail classes could be poor due to insufficient samples in learning. For the self-labeling-based methods, the head classes are often misclassified as the tail classes under the restrictive uniform distribution constraint. In addition, both types of methods tend to produce a biased classifier head induced by the long-tailed distribution setting. While recent work (Weng et al., 2021) proposes an unsupervised discovery strategy for long-tailed visual recognition, they mainly focus on the task of instance segmentation. By contrast, we systematically consider the problem of imbalanced novel class discovery in this work. Moreover, we propose an adaptive self-labeling learning framework based on the equiangular prototype representation and a novel optimal transport formulation, which enables us to mitigate the long-tail bias during class discovery in a principled manner.

## 2.2 Supervised Long-tailed Learning

There has been a large body of literature on supervised long-tailed learning (Zhang et al., 2021b), which aims to learn from labelled data with a long-tailed class distribution. To perform well on both head and tail classes, existing methods typically design a variety of strategies that improve the learning for the tail classes. One stream of strategies is to resample the imbalanced classes (Han et al., 2005) or to re-weight loss functions (Cao et al., 2019), followed by learning input representation and classifier head simultaneously. Along this line, Logit-Adjustment (LA) (Menon et al., 2020) is a simple and effective method that has been widely used, which mitigates the classifier bias by adjusting the logit based on class frequency during or after the learning. The other stream decouples the learning of input representation and classifier head (Kang et al., 2020; Zhang et al., 2021a). In particular, classifier retraining (cRT) (Kang et al., 2020) first learns a representation by instance-balanced resampling and then only retrains the classifier head with re-balancing techniques. Recently, inspired by the neural collapse (Papyan et al., 2020) phenomenon, Yang et al. (2022b) propose to initialize the classifier head as the vertices of a simplex equiangular tight frame (ETF), and fix its parameter during the learning, which helps to mitigate the classifier bias towards majority classes.

While those methods have achieved certain success in supervised long-tailed recognition, it is rather difficult to adopt them in novel class discovery as the class distribution of novel classes is unknown. Moreover, the problem of classifier bias becomes even more severe in discovering imbalanced novel data, where both the representation and classifier head are learned without ground-truth label information. To address this problem, we develop a novel representation learning strategy, which generalizes the ETF design and equal-sized optimal transport formulation, for modeling both long-tailed known and novel classes in a unified self-labeling framework.

# 3 Problem Setup and Method Overview

We consider the problem of Novel Class Discovery (NCD) for real-world visual recognition tasks, where the distributions of known and novel classes are typically long-tailed. In particular, we aim to learn a set of known classes $\mathcal{Y}^s$ from an annotated dataset $\mathcal{D}^s = \{(x_i^s, y_i^s)\}_{i=1}^N$, and to discover a set of novel classes $\mathcal{Y}^u$ from an unlabeled dataset $\mathcal{D}^u = \{x_i^u\}_{i=1}^M$. Here $x_i^s, x_i^u \in \mathcal{X}$ are the input images and $y_i^s$ are the known class labels in $\mathcal{D}^s$. For the NCD task, those two class sets have no overlap, i.e., $\mathcal{Y}^s \bigcap \mathcal{Y}^u = \emptyset$, and we denote

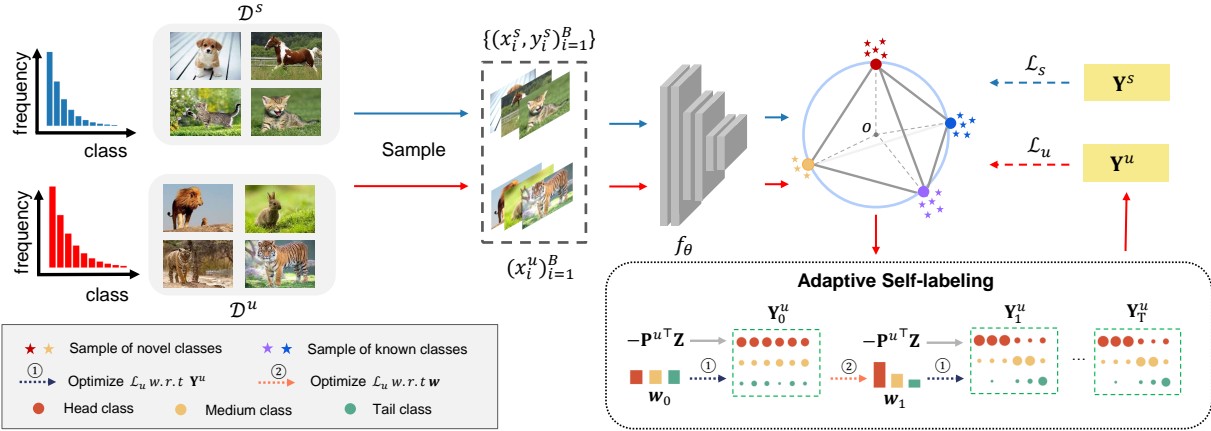

Figure 1: The overview of our framework. Our method first samples a data batch including known and novel classes from the long-tailed dataset and then encodes them into an embedding space. We adopt the equiangular prototypes for representing known and novel classes, and propose an adaptive self-labeling strategy to generate pseudo-labels for the novel classes. Our learning procedure alternates between pseudo-label generation, where we optimize $\mathcal{L}_u$ $w.r.t$ $\mathbf{Y}^u$, and minimizing an MSE loss, where we optimize $\mathcal{L}_u + \mathcal{L}_s$ $w.r.t$ $\mathbf{w}$. This process is repeated until convergence (See Sec.4.3 for details).

their sizes as $K^s$ and $K^u$ respectively. For long-tailed recognition, the numbers of training examples in different classes are severely imbalanced. For simplicity of notation, we assume that the known and novel classes are sorted by the cardinality of their training set in a descending order. Specifically, we denote the number of training data for the known class $i$ and the novel class $j$ as $N_i$ and $M_j$, accordingly, and we have $N_1 > N_2 \cdots > N_{K^s}, M_1 > M_2 \cdots > M_{K^u}$. To measure the class imbalance, we define an imbalance ratio for the known and novel classes, denoted as $R^s = \frac{N_1}{N_{K^s}}$ and $R^u = \frac{M_1}{M_{K^u}}$, respectively, and $R^s, R^u \gg 1$.

To tackle the NCD task for long-tailed recognition, we propose a novel adaptive self-labeling framework capable of better learning both known and novel visual classes under severe class imbalance. Our framework consists of three key ingredients that help alleviate the imbalance learning of known and novel classes: 1) We introduce a classifier design based on equiangular prototypes for both known and novel classes, which mitigates class bias due to its fixed parametrization; 2) For the novel classes, we develop a new adaptive self-labeling loss, which formulates the class discovery as a relaxed Optimal Transport problem and can be jointly optimized with the supervised loss of the known classes; 3) We design an efficient iterative learning algorithm that alternates between generating soft pseudo-labels for the novel-class data and performing representation learning. An overview of our framework is illustrated in Fig.1. In addition, we propose a simple method to estimate the number of novel class in the imbalance scenario. The details of our method will be introduced in Sec. 4.

## 4 Our Method

In this section, we first introduce our model architecture and class representations in Sec. 4.1, followed by the design of learning losses in Sec. 4.2 and Sec. 4.3 for the known and novel classes, respectively. Then we present our iterative self-labeling learning algorithm in Sec. 4.4. Finally, we provide the details of our strategy for estimating the number of novel classes under class imbalance scenario in Sec. 4.5.

### 4.1 Model Architecture and Class Representation

We tackle the long-tailed NCD problem by learning a deep network classifier for both known and novel classes. To this end, we adopt a generic design for the image classifier, consisting of an image encoder and a classification head for known and novel classes. Given an input $x$, our encoder network, denoted as $f_\theta$, computes a feature embedding $\mathbf{z} = f_\theta(x) \in \mathbb{R}^{D \times 1}$, which is then fed into the classification head for class prediction. Here we normalize the feature embedding such that $\|\mathbf{z}\|_2 = 1$. While any image encoder can

be potentially used in our framework, we adopt an unsupervised pre-trained ViT model (Dosovitskiy et al., 2021) as our initial encoder in this work, which can extract a discriminative representation robust to the imbalanced learning (Liu et al., 2022). We also share the encoder of known and novel classes to encourage knowledge transfer between two class sets during model learning.

For the classification head, we consider a prototype-based class representation where each class $i$ is represented by a unit vector $\mathbf{p}_i \in \mathbb{R}^{D \times 1}$. More specifically, we denote the class prototypes of the known classes as $\mathbf{P}^s = [\mathbf{p}_1^s, \cdots, \mathbf{p}_{K^s}^s]$ and those of the novel classes as $\mathbf{P}^u = [\mathbf{p}_1^u, \cdots, \mathbf{p}_{K^u}^u]$. The entire class space is then represented as $\mathbf{P} = [\mathbf{P}^s, \mathbf{P}^u] \in \mathbb{R}^{D \times (K^s + K^u)}$. To perform classification, given a feature embedding $\mathbf{z}$, we take the class of its nearest neighbor in the class prototypes $\mathbf{P}$ as follows,

$$c^* = \arg \min_i \ \|\mathbf{z} - \mathbf{P}_i\|_2, \tag{1}$$

where $\mathbf{P}_i$ is the $i$-th column of $\mathbf{P}$, and $c^*$ is the predicted class of the input $x$.

In the imbalanced class scenario, it is challenging to learn the prototypes from the data as they tend to bias toward the majority classes, especially for the novel classes, where the classifier and representation are learned without label information. While many calibration strategies have been developed for the long-tailed problems in supervised learning (c.f. Sec. 2), they are not applicable to the imbalanced novel class discovery task as the label distribution of novel classes is unknown.

To cope with imbalanced data in our NCD problem, our first design is to adopt a fixed parametrization for the class prototypes. Specifically, we generalize the strategy proposed by Yang et al. (2022b) for imbalanced supervised learning, and utilize the vertices of a simplex equiangular tight frame (ETF) as the prototype of both *known and novel* classes. By combining this prototypical representation with our adaptive self-labeling framework (Sec.4.3), our method is able to reduce the bias in learning for all the classes in a unified manner. More concretely, we generate the prototype set $\mathbf{P}$ as follows:

$$\mathbf{P} = \sqrt{\frac{K}{K-1}} \mathbf{M}(\mathbf{I}_K - \frac{1}{K}\mathbf{1}_{K \times K}), \tag{2}$$

where $\mathbf{M}$ is an arbitrary orthonormal matrix, $\mathbf{I}_K$ is a $K \times K$ identity matrix, $\mathbf{1}$ denotes the all ones matrix and $K = K^s + K^u$ is the total number of class prototypes. The generated prototypes have unit $l_2$ norm and same pair-wise angle, which treats all the classes equally and help debias the classifier learning.

## 4.2 Loss for Known Classes

For the known classes, we simply use the Mean Square Error (MSE) loss, which minimizes the $l_2$ distance between the feature embedding of an input $x_i^s$ and the class prototype of its groundtruth label $y_i^s$. Specifically, we adopt the average MSE loss on the subset of known classes $\mathcal{D}^s$ as follows,

$$\mathcal{L}_s(\theta) = \frac{1}{N}\sum_{i=1}^N \|\mathbf{z}_i^s - \mathbf{p}_{y_i^s}^s\|_2 = -\frac{1}{N}\sum_{i=1}^N 2\mathbf{z}_i^{s\top}\mathbf{p}_{y_i^s}^s + C, \tag{3}$$

where $\mathbf{z}_i^s, \mathbf{p}_{y_i^s}^s$ is the feature embeddings and class prototypes, respectively, $y_i^s$ is the ground-truth label of input $x_i^s$, and $C$ is a constant. It is worth noting that our design copes with the class imbalance in the known classes by adopting the equiangular prototype and initializing the encoder based on an unsupervised pre-trained network. This strategy is simple and effective (as shown in our experimental study)[1], and can be easily extended to discovering novel classes.

## 4.3 Adaptive Self-Labeling Loss for Novel Classes

We now present the loss function for discovering the novel classes in $\mathcal{D}^u$. Given an input $x_i^u$, we introduce a pseudo-label variable $y_i^u$ to indicate its (unknown) membership to the $K^u$ classes and define a clustering loss

---

[1]While it is possible to integrate additional label balancing techniques, it is beyond the scope of this work.

based on the Euclidean distance between its feature embedding $\mathbf{z}_i^u$ and the class prototypes $\mathbf{P}^u$ as follows,

$$\mathcal{L}_u(\theta) = \frac{1}{M}\sum_{i=1}^{M}\|\mathbf{z}_i^u - \mathbf{p}_{y_i^u}^u\|_2 = -\frac{1}{M}\sum_{i=1}^{M}2\mathbf{z}_i^{u\top}\mathbf{p}_{y_i^u}^u + C, \tag{4}$$

where $C$ is a constant as the feature and prototype vectors are normalized. Our goal is to jointly infer an optimal membership assignment and learn a discriminative representation that better discovers novel classes.

**Regularized Optimal Transport Formulation:** Directly optimizing $\mathcal{L}_u$ is difficult, and a naive alternating optimization strategy often suffers from poor local minima (Caron et al., 2018), especially under the scenario of long-tailed class distribution. To tackle this, we reformulate the loss in Eq. 4 into a regularized Optimal Transport (OT) problem, which enables us to design an adaptive self-labeling learning strategy that iteratively generates high-quality pseudo-labels (or class memberships) and optimizes the feature representation jointly with the known classes. To this end, we introduce two relaxation techniques to convert the Eq. 4 to an OT problem as detailed below.

First, we consider a soft label $\mathbf{y}_i^u \in \mathbb{R}_+^{1\times K^u}$ to encode the class membership of the datum $x_i^u$, where $\mathbf{y}_i^u\mathbf{1}_{K^u} = 1$. Ignoring the constants in $\mathcal{L}_u$, we can re-write the loss function in a vector form as follows,

$$\min_{\mathbf{Y}^u}\mathcal{L}_u(\mathbf{Y}^u;\theta) = \min_{\mathbf{Y}^u} -\frac{1}{M}\sum_{i=1}^{M}\langle\mathbf{y}_i^u, \mathbf{z}_i^{u\top}\mathbf{P}^u\rangle, \quad \text{s.t. } \mathbf{y}_i^u\mathbf{1}_{K^u} = 1 \tag{5}$$

$$= \min_{\mathbf{Y}^u}\langle\mathbf{Y}^u, -\mathbf{Z}^\top\mathbf{P}^u\rangle_F, \quad \text{s.t. } \mathbf{Y}^u\mathbf{1}_{K^u} = \boldsymbol{\mu} \tag{6}$$

where $\langle,\rangle_F$ represents the Frobenius product, $\mathbf{Y}^u = \frac{1}{M}[\mathbf{y}_1^u,\cdots,\mathbf{y}_M^u] \in \mathbb{R}_+^{M\times K^u}$ is the pseudo-label matrix, $\mathbf{Z} = [\mathbf{z}_1^u,\cdots,\mathbf{z}_M^u] \in \mathbb{R}^{D\times M}$ is the feature embedding matrix and $\boldsymbol{\mu} = \frac{1}{M}\mathbf{1}_M$. This soft label formulation is more robust to the noisy learning (Lukasik et al., 2020) and hence will facilitate the model learning with inferred pseudo-labels.

Second, inspired by (Asano et al., 2020), we introduce a constraint on the sizes of clusters to prevent degenerate solutions. Formally, we denote the cluster size distribution as a probability vector $\boldsymbol{\nu} \in \mathbb{R}_+^{K^u}$ and define the pseudo-label matrix constraint as $\mathbf{Y}^{u\top}\mathbf{1}_M = \boldsymbol{\nu}$. Previous methods typically take an equal-size assumption (Asano et al., 2020; Fini et al., 2021), where $\boldsymbol{\nu}$ is a uniform distribution. While such an assumption can partially alleviate the class bias by implicitly rebalancing the classes, it is often too restrictive for an unknown long-tailed class distribution. In particular, our preliminary empirical results show that it often forces the majority classes to be mis-clustered into minority classes, leading to noisy pseudo-label estimation. To remedy this, we propose a second relaxation mechanism on the above constraint. Specifically, we introduce an auxiliary variable $\mathbf{w} \in \mathbb{R}_+^{K^u}$, which is dynamically inferred during learning and encodes a proper constraint on the cluster-size distribution. More specifically, we formulate the loss into a regularized OT problem as follows:

$$\min_{\mathbf{Y}^u,\mathbf{w}}\mathcal{L}_u(\mathbf{Y}^u,\mathbf{w};\theta) = \min_{\mathbf{Y}^u,\mathbf{w}}\langle\mathbf{Y}^u, -\mathbf{Z}^\top\mathbf{P}^u\rangle_F + \gamma KL(\mathbf{w},\boldsymbol{\nu}), \tag{7}$$

$$\text{s.t.} \quad \mathbf{Y}^u \in \{\mathbf{Y}^u \in \mathbb{R}_+^{M\times K^u}|\mathbf{Y}^u\mathbf{1}_{K^u} = \boldsymbol{\mu}, \mathbf{Y}^{u\top}\mathbf{1}_M = \mathbf{w}\}, \tag{8}$$

where $\gamma$ is the balance factor to adjust the strength of KL constraint. When $\gamma = \inf$, the KL constraint falls back to equality constraints. Intuitively, our relaxed optimal transport formulation allows us to generate better pseudo labels adaptively, which alleviates the learning bias of head classes by proper label smoothing.

**Pseudo Label Generation:** Based on the regularized OT formulation of the clustering loss $\mathcal{L}_u$, we now present the pseudo label generation process when the encoder network $f_\theta$ is given. The generated pseudo labels will be used as the supervision for the novel classes, which is combined with the loss of known classes for updating the encoder network. We will defer the overall training strategy to Sec. 4.4 and first describe the pseudo label generation algorithm below.

Eq. (7) and (8) minimize $\mathcal{L}_u$ *w.r.t* $(\mathbf{Y}^u,\mathbf{w})$ with a fixed cost matrix $-\mathbf{Z}^\top\mathbf{P}^u$ (as $\theta$ is given), which can be solved by convex optimization techniques (Dvurechensky et al., 2018; Luo et al., 2023). However, they

---

**Algorithm 1:** Sinkhorn-Knopp Based Pseudo Labeling Algorithm

---

**Input:** Matrix $\mathbf{Y}$, marginal distribution $\boldsymbol{\mu}, \boldsymbol{\nu}$, hyper-parameters $T, \lambda$
**Output:** $\mathbf{Y}$
**Function** Pseudo-Labeling($\mathbf{Y}, \boldsymbol{\mu}, \mathbf{w}$):
    $\mathbf{Y} \leftarrow \exp(\mathbf{Y}/\lambda)$
    $\mathbf{Y} \leftarrow \mathbf{Y}/\sum \mathbf{Y}$
    $\alpha, \beta \leftarrow \mathbf{1}, \mathbf{1}$
    **for** $t \in 1, 2, .., T$ **do**
        $\alpha \leftarrow \boldsymbol{\mu}./(\mathbf{Y}\beta), \beta \leftarrow \mathbf{w}./(\mathbf{Y}^\top \alpha)$
    **end**
    $\mathbf{Y} \leftarrow diag(\alpha)\mathbf{Y}diag(\beta)$
    **return** $\mathbf{Y}$;
**End Function**

---

are typically computationally expensive in our scenario. Instead, we leverage the efficient Sinkhorn-Knopp algorithm (Cuturi, 2013) and propose a bi-level optimization algorithm to solve the problem approximately. Our approximate strategy consists of three main components as detailed below.

*A. Alternating Optimization with Gradient Truncation:* We adopt an alternating optimization strategy with truncated back-propagation (Shaban et al., 2019) to minimize the loss $\mathcal{L}_u(\mathbf{Y}^u, \mathbf{w})^2$. Specifically, we start from a fixed $\mathbf{w}$ (initialized by $\boldsymbol{\nu}$) and first minimize $\mathcal{L}_u(\mathbf{Y}^u, \mathbf{w})$ *w.r.t* $\mathbf{Y}^u$. As the KL constraint term remains constant, the task turns into a standard optimal transport problem, which can be efficiently solved by the Sinkhorn-Knopp Algorithm (Cuturi, 2013) as shown in Alg. 1. We truncate the iteration with a fixed $T$, which allows us to express the generated $\mathbf{Y}^u$ as a differentiable function of $\mathbf{w}$, denoted as $\mathbf{Y}^u(\mathbf{w})$. We then optimize $\mathcal{L}_u(\mathbf{Y}^u(\mathbf{w}), \mathbf{w})$ *w.r.t* $\mathbf{w}$ with simple gradient descent. The alternating optimization of $\mathbf{Y}^u$ and $\mathbf{w}$ takes several iterations to produce high-quality pseudo labels for the novel classes.

*B. Parametric Cluster Size Constraint:* Instead of representing the cluster size constraint $\mathbf{w}$ as a real-valued vector, we adopt a parametric function form in this work, which significantly reduces the search space of the optimization and typically leads to more stable optimization with better empirical results. Specifically, we parametrize $\mathbf{w}$ as a function of parameter $\tau$ in the follow form:

$$\mathbf{w}_i = \tau^{\frac{-i}{K^u-1}}, \quad i = 0, 1, ..., K^u - 1, \tag{9}$$

where $\tau$ can be viewed as the imbalance factor. As our class sizes decrease in our setting, we replace $\tau$ with a function form of $1 + \exp(\tau)$ in practice, which is always larger than 1. Then we normalize $\mathbf{w}_i$ by $\sum_{i=0}^{K^u-1} \mathbf{w}_i$ to make it a proper probability vector.

*C. Mini-Batch Buffer:* We typically generate pseudo labels in a mini-batch mode (c.f. Sec. 4.4), which however results in unstable optimization of the OT problem. This is mainly caused by poor estimation of the cost matrix due to insufficient data, especially in the long-tailed setting. To address this, we build a mini-batch buffer to store $J = 2048$ history predictions (i.e., $\mathbf{Z}^\top \mathbf{P}^u$) and replay the buffer to augment the batch-wise optimal transport computation. Empirically, we found that this mini-batch buffer significantly improves the performance of the novel classes.

## 4.4 Joint Model Learning

Given the loss functions $\mathcal{L}_s$ and $\mathcal{L}_u$, we now develop an iterative learning procedure for the entire model. As our classifier prototypes are fixed, our joint model learning focuses on the representation learning, parameterized by the encoder network $f_\theta$. Specifically, given the datasets of known and novel classes, $(\mathcal{D}^s, \mathcal{D}^u)$, we sample a mini-batch of known and novel classes data at each iteration, and perform the following two steps: 1) For novel-class data, we generate their pseudo labels by optimizing the regularized OT-based loss, as shown in Sec. 4.3; 2) Given the inferred pseudo labels for the novel-class data and the ground-truth labels for the known classes, we perform gradient descent on a combined loss function as follows,

$$\mathcal{L}(\theta) = \mathcal{L}_s(\theta) + \alpha\mathcal{L}_u(\theta), \tag{10}$$

---

[2]Note that we simplify $\mathcal{L}_u(\mathbf{Y}^u, \mathbf{w}; \theta)$ to $\mathcal{L}_u(\mathbf{Y}^u, \mathbf{w})$ as we do not optimize $\theta$ in pseudo label generation process.

---

**Algorithm 2:** Adaptive Self-labeling Algorithm

---

**Input:** $\mathcal{D}^s, \mathcal{D}^u$, encoder $f_\theta$, equiangular prototype $\mathbf{P} \in \mathbb{R}^{D \times (K^s + K^u)}$,
    initial mini-batch Buffer, $\mathbf{w}, \boldsymbol{\mu} = \mathbf{1}_{J \times 1}$, hyper-parameters $B, L$

**for** $e \in 1, 2, .., Epoch$ **do**
    **for** $s \in 1, 2, ..., Step$ **do**
        $\{(x_i^s, y_i^s)\}_{i=1}^B \leftarrow \text{Sample}(\mathcal{D}^s), \{x_i^u\}_{i=1}^B \leftarrow \text{Sample}(\mathcal{D}^u)$
        $\mathbf{z}^s = f_\theta(x^s), \mathbf{z}^u = f_\theta(x^u)$
        `//MSE loss for labeled data`
        $\mathcal{L}_s = \frac{1}{B} \sum_{i=1}^B ||\mathbf{z}_i^s - \mathbf{P}_{y_i^s}||^2$
        $\mathbf{y}^u = \mathbf{z}^{u\top} \mathbf{P}^u \in \mathbb{R}^{1 \times K^u}$
        $\mathbf{Y}^u = \text{Buffer}([\mathbf{y}_1^u; \mathbf{y}_2^u ..; \mathbf{y}_M^u]) \in \mathbb{R}^{J \times K^u}$
        **for** $l \in 1, 2, ..., L$ **do**
            $\mathbf{Y}^u = \text{Pseudo-Labeling}(\mathbf{Y}^u, \boldsymbol{\mu}, \mathbf{w})$
            $\mathbf{w} \approx \arg\min_{\mathbf{w}} \mathcal{L}_u(\mathbf{Y}^u(\mathbf{w}), \mathbf{w})$
        **end**
        `//MSE loss for unlabeled data`
        $\mathcal{L}_u = \langle \mathbf{Y}^u, -\mathbf{Z}^\top \mathbf{P}^u \rangle_F$
        minimize $\mathcal{L}_s + \alpha \mathcal{L}_u$ $w.r.t$ $\theta$
    **end**
**end**

---

where $\mathcal{L}_s$ and $\mathcal{L}_u$ are the losses for the known and novel classes respectively, and $\alpha$ is the weight to balance the learning of known and novel classes. The above learning process minimizes the overall loss function over the encoder parameters and pseudo labels in an alternative manner until converge. An overview of our entire learning algorithm is illustrated in Alg.2.

## 4.5 Estimating the Number of Novel Classes

In the scenarios with unknown number of novel classes, we introduce a simple and effective strategy for estimating the cardinality of the imbalanced novel class set, $K^u$. To achieve this, our method utilizes the data clustering of the known classes to score and select the optimal $K^u$ (Vaze et al., 2022). Specifically, given a candidate $K^u$, we first perform a hierarchical clustering algorithm to cluster both known and novel classes ($\mathcal{D}^s, \mathcal{D}^u$) into $K^u$ clusters. Next, we employ the Hungarian algorithm to find the optimal mapping between the set of cluster indices and known class labels of data, and evaluate the clustering accuracy of the known classes. To determine the best value for $K^u$, we repeat this process for a set of candidate sizes and choose the setting with the highest accuracy of the known classes.

However, in imbalanced datasets, the average performance of known classes tends to be biased towards larger classes, which results in underestimation of the number of unknown classes. To address this, we consider the average accuracy over classes, which is less influenced by class imbalance, and design a mixed metric for selecting the optimal value of $K^u$. Concretely, the mixed metric is defined as a weighted sum of the overall accuracy of the known classes, denoted by $Acc_s$, and the average class accuracy of known classes, denoted by $Acc_c$, as follows:

$$Acc = \beta Acc_s + (1 - \beta) Acc_c, \tag{11}$$

where $\beta$ is a weighting parameter and is set as 0.5 empirically. We employ the mixed metric to perform a binary search for the optimal value of $K^u$. The detail algorithm is shown in Appendix A.

# 5 Experiments

## 5.1 Experimental Setup

**Datasets** We evaluate the performance of our method on four datasets, including long-tailed variants of two image classification datasets, CIFAR100 (Krizhevsky et al., 2009) and ImageNet100 (Deng et al., 2009), and two real-world medium/large-scale long-tailed image classification datasets, Herbarium19 (Tan et al., 2019) and iNaturalist18 (Van Horn et al., 2018). For the iNaturalist18 dataset, we subsample 1000 and

Table 1: The details of all the datasets for evaluation. The imbalance ratio of the known classes $R^s$ is 50 for CIFAR100-50-50 and ImageNet100-50-50, and $R^u$ denotes the imbalance ratio of the novel classes.

| Datasets $R^u$ | CIFAR100-50-50 50 | 100 | ImageNet100-50-50 50 | 100 | Herbarium19 UnKnown | iNaturalist18-1K UnKnown | iNaturalist18-2K UnKnown |
|---|---|---|---|---|---|---|---|
| Known Classes | 50 | 50 | 50 | 50 | 342 | 500 | 1000 |
| Known Data | 6.4k | 6.4k | 16.5k | 16.5k | 17.8k | 26.3k | 52.7k |
| Novel Classes | 50 | 50 | 50 | 50 | 342 | 500 | 1000 |
| Novel Data | 6.4k | 5.5k | 16.2k | 14.0k | 16.5k | 26.4k | 49.9k |
| Test set | 10k | 10k | 5.0k | 5.0k | 2.8k | 3.0k | 6.0k |

2000 classes from iNaturalist18 to create iNaturalist18-1K and iNaturalist18-2K, respectively, which allows us to test our algorithm on a relatively large-scale benchmark with a practical computation budget. For each dataset, we randomly divide all classes into 50% known classes and 50% novel classes. For CIFAR100 and ImageNet100, we create "long-tailed" variants of the known and novel classes by downsampling data examples per class following the exponential profile in (Cui et al., 2019) with imbalance ratio $R = \frac{N_1}{N_K}$. In order to explore the performance of novel class discovery under different settings, we set the imbalance ratio of known classes $R^s$ as 50 and that of novel classes $R^u$ as 50 and 100, which aim to mimic typical settings in long-tailed image recognition tasks. For testing, we report the NCD performance on the official validation sets of each dataset, except for CIFAR100, where we use its official test set. We note that those test sets have uniform class distributions, with the exception of the Herbarium19 dataset, and encompass both known and novel classes. The details of our datasets are shown in Tab. 1.

**Metric**  To evaluate the performance of our model on each dataset, we calculate the average accuracy over classes on the corresponding test set. We measure the clustering accuracy by comparing the hidden ground-truth labels $y_i$ with the model predictions $\hat{y}_i$ using the following strategy:

$$\texttt{ClusterAcc} = \max_{perm \in P} \frac{1}{N} \sum_{i=1}^{N} \mathbb{1}(y_i = perm(\hat{y}_i)), \tag{12}$$

where $P$ represents the set of all possible permutations of test data. To compute this metric, we use the Hungarian algorithm (Kuhn, 1955) to find the optimal matching between the ground-truth and prediction. We note that we perform the Hungarian matching for the data from all categories, and then measure the classification accuracy on both the known and novel subsets. We also sort the categories according to their sizes in a descending order and partition them into 'Head', 'Medium', 'Tail' subgroups with a ratio of 3: 4: 3 (in number of classes) for all datasets.

**Implementation Details**  For a fair comparison, all the methods in evaluation use a ViT-B-16 backbone network as the image encoder, which is pre-trained with DINO (Caron et al., 2021) in an unsupervised manner. For our method, we train 50 epochs on CIFAR100 and ImageNet100, 70 epochs on Herbarium and iNaturalist18. We use AdamW with momentum as the optimizer with linear warm-up and cosine annealing ($lr_{base}$ = 1e-3, $lr_{min}$ = 1e-4, and weight decay 5e-4). We set $\alpha = 1$, and select $\gamma = 500$ by validation. In addition, we analyze the sensitivity of $\gamma$ in Appendix C. For all the experiments, we set the batch size to 128 and the iteration step $L$ to 10. For the Sinkhorn-Knopp algorithm, we adopt all the hyperparameters from (Caron et al., 2020), e.g. $n_{iter} = 3$ and $\epsilon = 0.05$. Implementation details of other methods can be found in Appendix B.

## 5.2   Comparison with NCD Baselines

We first report a comparison of our method with two major NCD baselines (Autonovel and UNO) and their variants on the benchmarks, CIFAR100 and ImageNet100 in Tab.2. In addition to the original NCD algorithms, we also apply two long-tailed learning techniques to the baselines, which include logits adjustment (LA), which adjusts the logits according to the estimated class distribution in the inference, and class reweighting (cRT), which retrains the classifier with a class-balanced dataset. For CIFAR100, when $R^s =$

Table 2: Long-tailed novel class discovery performance on CIFAR-100, ImageNet100. We report average class accuracy on test datasets. $R^s, R^u$ are the imbalance factors of known and novel classes respectively. "+LA" means post-processing with logits-adjustment (Menon et al., 2020), "+cRT" means classifier retraining (Kang et al., 2020).

| | CIFAR100-50-50 | | | | | | ImageNet100-50-50 | | | | | |
| | $R^s = 50, R^u = 50$ | | | $R^s = 50, R^u = 100$ | | | $R^s = 50, R^u = 50$ | | | $R^s = 50, R^u = 100$ | | |
| Method | All | Novel | Known | All | Novel | Known | All | Novel | Known | All | Novel | Known |
|---|---|---|---|---|---|---|---|---|---|---|---|---|
| Autonovel | 44.42 | 22.28 | 66.56 | 44.04 | 22.12 | 65.96 | 67.50 | 47.96 | 87.04 | 63.86 | 40.88 | 86.84 |
| Autonovel + LA | 45.32 | 20.76 | 69.88 | 42.20 | 18.00 | 66.40 | 67.74 | 46.72 | **88.76** | 64.08 | 39.32 | **88.84** |
| AutoNovel + cRT | 47.20 | 26.48 | 67.92 | 41.94 | 22.62 | 61.26 | 67.76 | 49.88 | 85.64 | 63.90 | 42.40 | 85.40 |
| UNO | 50.82 | 34.10 | 67.54 | 49.50 | 31.24 | 67.76 | 65.30 | 43.08 | 87.52 | 62.52 | 37.84 | 87.20 |
| UNO + LA | 52.36 | 33.82 | **70.90** | 51.62 | 31.04 | **72.20** | 65.92 | 43.12 | 88.72 | 63.28 | 37.72 | 88.84 |
| UNO + cRT | **54.26** | 40.42 | 68.10 | 47.62 | 31.02 | 64.22 | 68.38 | 50.80 | 85.96 | 63.10 | 39.96 | 86.24 |
| Ours | 53.75 | **40.60** | 66.90 | **51.90** | **36.80** | 67.00 | **73.94** | **61.48** | 86.40 | **69.38** | **51.96** | 86.80 |

Table 3: Long-tailed novel class discovery performance on medium/large-scale Herbarium19 and iNaturalist18. Other details are the same as Tab. 2.

| | Herbarium | | | iNaturalist18-1K | | | iNaturalist18-2K | | |
| Method | All | Novel | Known | All | Novel | Known | All | Novel | Known |
|---|---|---|---|---|---|---|---|---|---|
| Autonovel | 34.58 | 9.96 | 59.30 | 42.33 | 11.67 | 73.00 | 39.08 | 8.57 | 69.60 |
| Autonovel + LA | 32.54 | 8.60 | 56.56 | 42.40 | 11.27 | **73.53** | 44.67 | 14.33 | **75.00** |
| AutoNovel + cRT | 45.05 | 24.46 | 64.49 | 44.20 | 16.13 | 72.27 | 37.95 | 9.27 | 66.63 |
| UNO | 47.47 | 34.50 | 60.58 | 52.93 | 31.60 | 74.27 | 45.60 | 19.97 | 71.23 |
| UNO + LA | 46.76 | 27.96 | **65.69** | 46.63 | 24.33 | **74.60** | 46.63 | 20.33 | 72.93 |
| UNO + cRT | 46.47 | 33.13 | 59.95 | 51.73 | 32.60 | 70.87 | 46.47 | 24.90 | 68.03 |
| Ours | **49.21** | **36.93** | 61.63 | **58.87** | **45.47** | 72.27 | **49.57** | **34.13** | 65.00 |

$R^u = 50$, our method achieves competitive results compared to the two-stage training methods. As the data become more imbalanced, i.e. $R^u = 100$, our method achieves **5.78**% improvement on the novel class accuracy. Here we note that our method does not exhibit a significant advantage due to the limited quality of the representation computed from the low-resolution images. For ImageNet100, our method achieves significant improvements in two different $R^u$ settings, surpassing the previous SOTA method by **10.68**% and **9.56**% respectively.

Furthermore, in Tab. 3, we show the results on two medium/large scale datasets. Specifically, on the challenging fine-grained imbalanced Herbarium19 dataset, which contains 341 known classes, our method achieves **2.43**% improvement on the novel class accuracy compared to UNO. We also report the per-sample average class accuracy in Appendix E, on which we achieve $\sim 10\%$ improvement. On the more challenging iNaturalist18 datasets, we observe a significant improvement (>**10**%) in the performance of novel classes compared to the Herbarium19 dataset. In summary, our notable performance gains in multiple experimental settings demonstrate that our method is effective for the challenging long-tailed NCD task.

## 5.3 NCD with Unknown Number of Novel Categories

To evaluate the effectiveness of our estimation method, we establish a baseline that uses the average accuracy as the indicator to search for the optimal $K^u$ value by hierarchical clustering. The details of the baseline algorithm are shown in Appendix A. As shown in Tab.4, our proposed method significantly outperforms the baseline on three datasets and various scenarios, indicating the superiority of our proposed mixed metric for estimating the number of novel classes in imbalanced scenarios.

Furthermore, we conduct experiments on three datasets with estimated $K^u$. As Tab.5 shown, our method achieves sizeable improvements on the novel and overall class accuracy, except in the case of CIFAR when $R^u = 50$, which we achieve comparable results. On the CIFAR dataset, the existing methods outperform our method on the known classes, especially when the LA or cRT techniques are integrated. As a result, our method demonstrates only slight improvements or comparable results w.r.t the existing methods on the overall accuracy. However, it is worth noting that when our method is combined with the LA technique, we

Table 4: Estimation the number of novel categories. $R^s$ is 50 for CIFAR100 and ImageNet100 datasets.

| Method | CIFAR100-50-50 | | ImageNet100-50-50 | | Herbarium |
|--------|-----------------|-----------------|-------------------|-------------------|-----------|
|        | $R^u = 50$ | $R^u = 100$ | $R^u = 50$ | $R^u = 100$ | Unknown |
| GT       | 50 | 50 | 50 | 50 | 341 |
| Baseline | 0  | 10 | 7  | 14 | 2   |
| Ours     | 20 | 29 | 59 | 59 | 153 |

Table 5: Experiments on three datasets when $K^u$ is unknown.

| | **CIFAR100-50-50** | | | | | | **ImageNet100-50-50** | | | | | | **Herbarium** | | |
|--|--|--|--|--|--|--|--|--|--|--|--|--|--|--|--|
| | $R^s = 50, R^u = 50$ | | | $R^s = 50, R^u = 100$ | | | $R^s = 50, R^u = 50$ | | | $R^s = 50, R^u = 100$ | | | **Unknown** | | |
| Method | All | Novel | Known | All | Novel | Known | All | Novel | Known | All | Novel | Known | All | Novel | Known |
| AutoNovel     | 41.25 | 16.08 | 66.42 | 43.74 | 17.64 | **69.84** | 63.92 | 37.80 | 90.04 | 66.68 | 44.96 | 88.40 | 37.84 | 15.64 | 60.16 |
| AutoNovel+LA  | 41.82 | 16.18 | 67.46 | 43.82 | 18.08 | 69.56 | 61.74 | 32.68 | **90.80** | 62.26 | 35.52 | **89.00** | 42.15 | 19.06 | **65.37** |
| AutoNovel+cRT | 45.81 | 19.50 | **72.12** | 43.44 | 18.18 | 68.70 | 62.83 | 37.96 | 87.68 | 52.44 | 16.68 | 88.20 | 40.95 | 18.86 | 63.16 |
| UNO           | 47.67 | 28.92 | 66.42 | 46.56 | 25.92 | 67.20 | 67.96 | 48.48 | 87.44 | 64.16 | 41.24 | 87.08 | 40.83 | 23.55 | 58.24 |
| UNO+LA        | 49.51 | 28.18 | 70.84 | 48.02 | 25.70 | 70.34 | 67.94 | 47.44 | 88.44 | 65.02 | 41.48 | 88.56 | 42.83 | 23.02 | 62.56 |
| UNO+cRT       | 49.35 | 30.82 | 67.88 | 45.49 | 26.68 | 64.30 | 70.94 | 55.32 | 86.56 | 62.76 | 38.72 | 86.80 | 40.67 | 22.84 | 58.63 |
| Ours          | 49.03 | **32.66** | 65.40 | 48.89 | 33.06 | 64.72 | 74.06 | 61.04 | 87.08 | 68.94 | 50.84 | 87.04 | 44.20 | 29.01 | 59.34 |
| Ours+LA       | **49.64** | 32.46 | 66.82 | **49.87** | **33.12** | **66.62** | **74.38** | **61.48** | 87.28 | **71.08** | **54.92** | 87.24 | **45.74** | **29.55** | 61.90 |

achieve more favorable outcomes. On the ImageNet100 and Herbarium19 datasets, our method surpasses the existing methods by a large margin. For example, on ImageNet100 dataset when $R^u = 100$, ours outperforms the best baseline (AutoNovel) by 3.92% in overall accuracy and 5.88% in novel-class accuracy. Moreover, when our method is equipped with LA, the performance is further improved, with an increase of 4.4% in overall accuracy and 9.96% in novel-class accuracy.

It is important to note that the effect of estimated $K^u$ differs based on its relationship with the ground truth value. When the estimated $K^u$ is lower than the ground truth value, such as CIFAR100 and Herbarium19, the performance deteriorates compared to using the true $K^u$. This occurs because the estimated lower $K^u$ leads to the mixing of some classes, especially medium and tail classes, resulting in degraded performance. When the estimated $K^u$ is higher than the ground truth value, such as ImageNet100, using the estimated $K^u$ leads to better results for UNO and comparable results for Ours. For UNO which assume equally sized distribution, larger $K^u$ tend to assign the head classes to additional empty classes, reducing the noise caused by the mixing of head classes with medium and tail classes, and thereby improving the accuracy of medium and tail classes. By contrast, our method dynamically adjusts the allocation ratio for novel classes, effectively suppressing the assignment of head classes to empty classes, which allows us to achieve comparable results. A more detailed analysis of this phenomenon can be found in Appendix F.

## 5.4 Ablation Study

**Component Analysis:** Table 6 presents ablation results of our method's components on ImageNet100. The impact of each core component is analyzed individually, including equiangular prototype representation, adaptive self-labeling, and the mini-batch buffer. As shown in the first and second rows of the Tab.6, the addition of mini-batch buffer results in a 2% improvement compared to the baseline on novel class accuracy. By comparing the second and third rows, notable improvements emerge in subgroup class accuracy, especially for head classes. For instance, with $R^u = 100$, our method achieves 16.67% improvement on the head, 7.8% improvement on medium, 7.33% improvement on tail classes. This demonstrates that the equiangular prototype representation helps alleviate the imbalance learning of novel classes. Comparing the third and last row, we show that adopting adaptive self-labeling greatly improves the tail and head class accuracy. For example, our method achieves 11.47% improvement on head, and 7.34% improvement on tail classes for $R^u = 50$. Such results indicate that the uniform constraint on the distribution of clusters is not suitable for imbalance clustering, as it tends to misclassify head classes samples as tail classes. And the reason for the slightly worse performance for medium classes is that the uniform distribution constraint better approximates the true medium class distribution. However, this constraint harms the performance of the head and tail, resulting in a nearly 4% decrease in the novel class accuracy. In conclusion, the overall results validate the

Table 6: Ablation on ImageNet100. "EP" and "ASL" stands for equiangular prototype and adaptive self-labeling.

| Method | $R^s = 50, R^u = 50$ | | | | $R^s = 50, R^u = 100$ | | | |
|---|---|---|---|---|---|---|---|---|
| | Novel | Head | Medium | Tail | Novel | Head | Medium | Tail |
| Baseline | 43.08 | 44.93 | 49.20 | 33.07 | 37.84 | 49.73 | 43.10 | 18.93 |
| + Buffer | 46.36 | 46.93 | 51.90 | 38.40 | 39.88 | 52.00 | 46.00 | 19.60 |
| + Buffer + EP | 57.40 | 66.00 | **59.70** | 45.73 | 47.24 | 68.67 | 49.20 | 23.20 |
| **+ Buffer + EP + ASL** | **61.48** | **77.47** | 55.80 | **53.07** | **51.96** | **77.47** | **54.20** | **23.47** |

Table 7: The effects of different combinations of loss function and classifier. Results on ImageNet100, for $R^s = 50, R^u = 50$. cls is an abbreviation for classifier.

| Method | Novel | Head | Medium | Tail |
|---|---|---|---|---|
| Learnable cls + CE loss | 46.36 | 46.93 | 51.90 | 38.40 |
| EP cls + CE loss | 52.40 | 53.47 | **66.10** | 33.07 |
| **EP cls + MSE loss** | **57.40** | **66.00** | 59.70 | **45.73** |

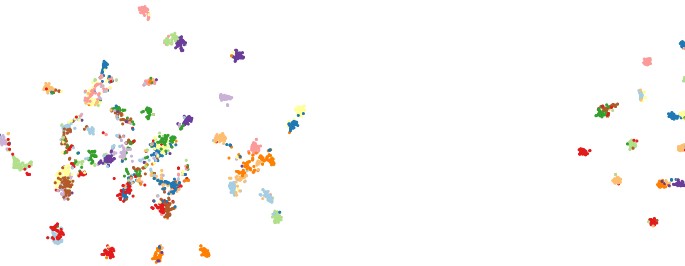

Figure 2: t-SNE visualization of novel instances in ImageNet100 for features after the last transform block. The left is the feature space using a learnable classifier, and the right is the feature space using equiangular prototype.

effectiveness of our proposed components, and especially the equiangular prototype and adaptive self-labeling produce notable improvements.

**Effect of Equiangular Prototype:** In Sec.4, we utilize the MSE loss to minimize the distance between sample and prototype. Here we delve into the impacts of various loss function and classifier combinations. As Tab.7 shown, a learnable classifier with CE loss performs worse than EP cls + CE loss. This discrepancy can be attributed to the tendency of the learned prototypes exhibiting bias towards head classes, ultimately leading to less discriminative representations. What's more, EP cls with MSE loss improve EP cls with CE loss by a large margin, especially for head and tail classes. In the early learning stage, the representation is relatively poor, and massive head class samples are allocated to tail classes because of uniform distribution constraints. While the gradient of CE loss is larger than MSE loss, resulting in the EP + CE loss overfitting to the noise pseudo-label quickly.

Moreover, to better understand the effect of the equiangular prototype for novel class clustering, we visualize the feature space by t-SNE(Van der Maaten & Hinton, 2008) on the test set. As Fig.2 shown, the features learned by the equiangular prototype are more tightly grouped with more evenly distributed interclass distances. However, the learnable classifier setting results in several classes being entangled together.

**Adaptive Self-labeling:** In this part, we validate the effectiveness of our design on **w**. We set **w** as the uniform and ground-truth distribution, and conduct corresponding experiments. Interestingly, as shown in the first two rows of Tab.8, setting w as a uniform prior achieves better performance, especially on medium and tail class. We speculate that the uniform constraint smooths the pseudo label of head class samples, mitigating the bias learning of head classes, thus improving the results of medium and tail classes. In addition, we also try two ways to learn **w** with a prior constraint of uniform distribution. One way is to parameterize w as a real-valued vector, and the other is to use a parametric form as a function of $\tau$. As shown in the last two rows of Tab.8, we find that optimizing a k-dimensional **w** is unstable and prone to

Table 8: The impact of different parametric strategy of **w**. The first two rows of **w** are fixed, and the last two rows represent the two parameterization ways of **w**. Results on ImageNet100, for $R^s = 50, R^u = 50$.

| Method | Novel | Head | Medium | Tail |
|---|---|---|---|---|
| $\mathbf{w}$ = Uniform | 57.40 | 66.00 | 59.70 | 45.73 |
| $\mathbf{w}$ = True Prior | 42.40 | 64.53 | 46.90 | 14.27 |
| $\mathbf{w}$ | 55.64 | 69.73 | **59.80** | 31.87 |
| $\mathbf{w}(\tau)$ | **61.48** | **77.47** | 55.80 | **53.07** |

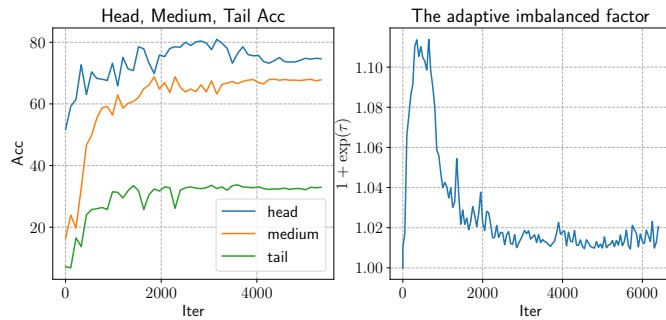

Figure 3: Analysis of **w** during training

assign overly large cluster sizes for some clusters. Therefore, optimizing **w** parametrized by a function of parameter $\tau$ (As shown in Eq.9) seems to be more effective.

To provide more analysis on **w**, we visualize the learned imbalance factor and the head/medium/tail class accuracy during the training process of the model. As Fig.3 shows, in the early stage, the head class accuracy first increases quickly, indicating that the model bias to the head classes and their representations are better learned. Correspondingly, the learned imbalance factor increases quickly, thus assigning more samples to the head classes. Subsequently, the medium and tail class accuracy increase; meanwhile, the imbalance factor decreases, resulting in the pseudo label process biasing to the medium and tail classes. Although the imbalance factor changes only slightly during the learning, it improves novel class accuracy by a large margin compared to the fixed uniform prior (the first row and last row in Tab.8).

## 6 Conclusion

In this paper, we present a real-world novel class discovery setting for visual recognition, where known and novel classes have long-tailed distributions. To mitigate the impact of imbalance on learning all classes, we have proposed a unified adaptive self-labeling framework, which introduces an equiangular prototype-based class representation and an iterative pseudo-label generation strategy for visual class learning. In particular, we formulate the pseudo-label generation step as a relaxed optimal transport problem and develop a bi-level optimization algorithm to efficiently solve the problem. Moreover, we propose an effective method to estimate the number of novel classes in the imbalanced scenario. Finally, we validate our method with extensive experiments on two small-scale long-tailed datasets, CIFAR100 and ImageNet100, and two medium/large-scale real-world datasets, Herbarium19 and iNaturalist18. Our framework achieves competitive or superior performances, demonstrating its efficacy.

## Acknowledgement

This work was supported by Shanghai Science and Technology Program 21010502700, Shanghai Frontiers Science Center of Human-centered Artificial Intelligence and the MoE Key Laboratory of Intelligent Perception and Human-Machine Collaboration (ShanghaiTech University).

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

## A   The algorithm of estimating the number of novel categories

We introduce a novel approach for estimating the number of unknown classes in imbalanced scenarios (Section 4.5). Our method leverages the clustering performance of the known classes dataset $\mathcal{D}^s$ as a means of searching for the optimal value of $K^u$. The detailed algorithm for this estimation process is outlined in Algorithm 3. In our method, we evaluate the performance using the mixed metric (Eqn. 11). In contrast, the "Baseline" method utilizes the average accuracy of each sample as its evaluation metric, which tends to bias to majority classes.

---

**Algorithm 3:** The algorithm of estimating the number of novel categories.

---

**Input:** $\mathcal{D}^s, \mathcal{D}^u, K^s$, maximum $K^u_{max}$, evaluation metric *eval*, hierarchical clustering algorithm $HC$
**Output:** $K^u_{mid}$
$K^u_{high}, K^u_{med}, K^u_{low} \leftarrow K^u_{max}, K^u_{max}//2, 0$
$Acc_{high} = eval(HC(\mathcal{D}^s, \mathcal{D}^u, K^u_{high} + K^s), \mathcal{D}^s)$
$Acc_{med} = eval(HC(\mathcal{D}^s, \mathcal{D}^u, K^u_{med} + K^s), \mathcal{D}^s)$
$Acc_{low} = eval(HC(\mathcal{D}^s, \mathcal{D}^u, K^u_{low} + K^s), \mathcal{D}^s)$
**while** $K^u_{high} > K^u_{low}$ **do**
    **if** $Acc_{high} > Acc_{low}$ **then**
        $K^u_{low} \leftarrow K^u_{med}$
        $K^u_{med} \leftarrow (K^u_{high} + K^u_{low})//2$
        $Acc_{low} = Acc_{med}$
        $Acc_{med} = eval(HC(\mathcal{D}^s, \mathcal{D}^u, K^u_{med} + K^s), \mathcal{D}^s)$
    **end**
    **else**
        $K^u_{high} \leftarrow K^u_{med}$
        $K^u_{med} \leftarrow (K^u_{high} + K^u_{low})//2$
        $Acc_{high} = Acc_{med}$
        $Acc_{med} = eval(HC(\mathcal{D}^s, \mathcal{D}^u, K^u_{med} + K^s), \mathcal{D}^s)$
    **end**
**end**

---

## B   Implementation details

As there are currently no existing baselines for novel class discovery in an imbalanced setting, we have implemented two typical NCD methods, AutoNovel (Han et al., 2021) and UNO (Fini et al., 2021). To handle imbalanced learning, we have combined these NCD methods with two common approaches for long-tailed problems: logit-adjustment (Menon et al., 2020) and decoupling the learning of representation and classifier head (Kang et al., 2020).

We have used the same unsupervised pretrained model and only modified the training setup of AutoNovel and UNO. Specifically, we trained AutoNovel for 200 epochs until convergence on all datasets, and the training strategy for UNO is identical to ours, as described in the main paper.

For our implementation of logit-adjustment, we have set $\pi = 1$ following (Menon et al., 2020). If the estimated number of pseudo-labels for a novel class is 0, we do not make any corrections to its logits. When adding cRT (Kang et al., 2020), we first estimate the class distribution and use the same number of epochs as in the first stage.

## C   Sensitive analysis of $\gamma$

The optimal value for the hyperparameter $\gamma$ is selected by partitioning a subset of known classes as the validation set. Additionally, to investigate the sensitivity of the hyperparameter gamma, we have presented the change in novel class accuracy from gamma values of 100 to infinity in Figure 4. Our findings indicate that when gamma is larger than 300, our adaptive self-labeling method outperforms the naive baseline. However, the value selected using our validation set is not the optimal one.

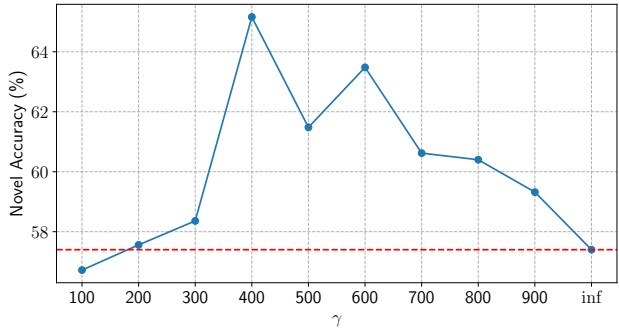

Figure 4: The sensitive analysis of $\gamma$

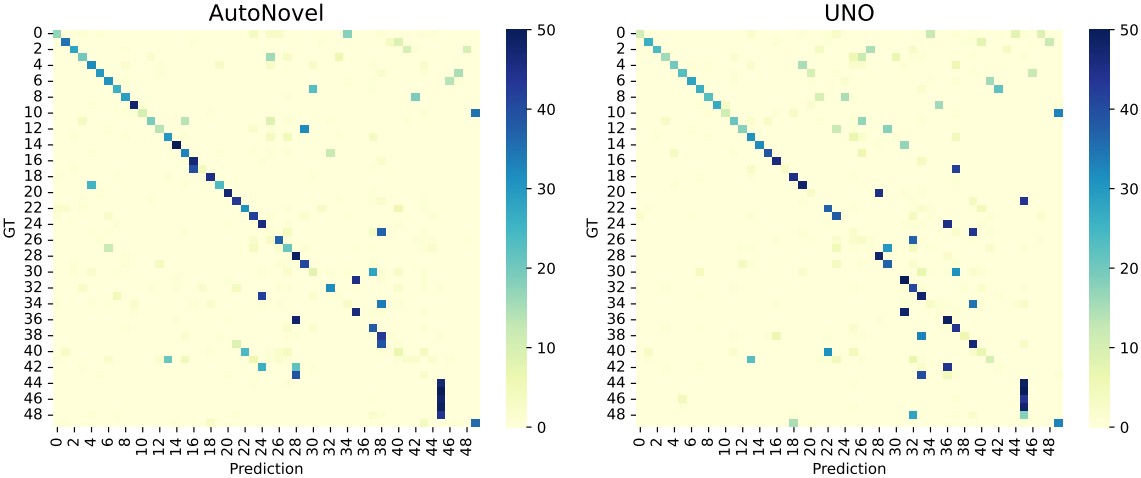

Figure 5: The confusion matrix of novel classes for typical NCD methods.

Table 9: The details analysis of typical NCD methods. Results on ImageNet100, for $R^s = 50, R^u = 50$.

| Method | Novel | Head | Medium | Tail |
|---|---|---|---|---|
| Autonovel | 47.96 | 55.87 | 55.60 | 29.87 |
| UNO | 43.08 | 44.93 | 49.50 | 32.13 |
| Ours | **61.48** | **77.47** | **55.80** | **53.07** |

Table 10: The results of UNO on the balance dataset.

| Dataset | All | Novel | Known |
|---|---|---|---|
| CIFAR100 | 74.55 | 65.40 | 83.70 |
| ImageNet100 | 85.12 | 76.96 | 93.28 |

# D   Analysis of NCD method

aragraphNCD methods on Head/Medium/Tail classes: In Tab. 2 of the main manuscripts, we have presented the results of NCD methods. To further analyze these methods, we have shown their performance on the Head, Medium, and Tail classes in the novel class in Tab. 9. Our proposed method shows an improvement of over 20% on both the Head and Tail classes, demonstrating its advantage.

Additionally, Autonovel performs worse on the Tail class due to the limited number of positive pair samples for tail classes. In contrast, UNO performs worse in the Head classes because the head classes are misclassified into Tail classes. This argument is supported by the confusion matrix shown in Fig. 5. Specifically, in the case of Autonovel, several tail classes are merged into a single class due to poor representation. There are too many samples on the right side of the confusion matrix for UNO, which denotes that the head classes are being misclassified into tail classes.

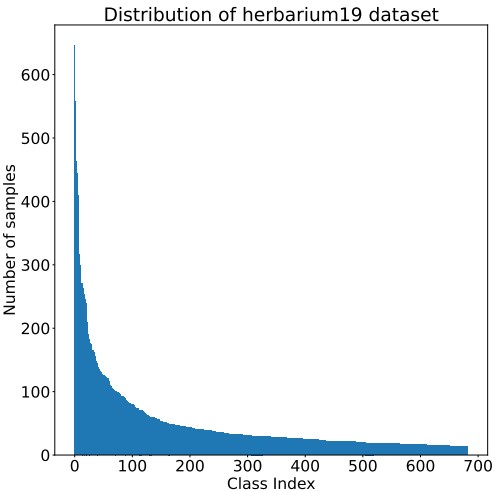

Figure 6: The distribution of herbarium19 dataset

**NCD methods with class re-balancing techniques:**  We conclude that novel class discovery (NCD) for long-tailed data is challenging, and existing methods have not been able to solve this problem. As shown in Tab. 2 of the main paper and Tab.10, the novel class accuracy decreased by almost 30% on both CIFAR100 and ImageNet100 datasets.

To improve the performance of NCD in long-tailed scenarios, we have combined NCD with long-tail methods (+LA, +cRT). We observe that the accuracy of known and novel classes improves when the distribution estimation of novel classes is more accurate. Specifically, in CIFAR100, when $R^u = 50$, AutoNovel performs poorly in estimating the distribution of novel classes due to the use of pairwise loss, which assigns similar features the same pseudo label, making it difficult to learn distinctive representations for tail classes in an imbalanced setting. This results in tail classes being mixed with head classes. When AutoNovel is combined with long-tail methods, the novel classes decrease while UNO improves. When $R^u = 100$, the severe imbalance of novel classes makes learning novel classes more difficult, resulting in a worse estimated distribution. As a result, combining UNO with long-tail methods no longer has any effect.

On ImageNet, the estimated distribution of novel classes is more accurate, and both AutoNovel and UNO have improved the accuracy of novel class. However, UNO's accuracy of known classes slightly decreased because novel classes and known classes are often confused. For Herbarium19, the actual distribution is difficult to predict, so the achievement of LA and cRT is limited.

In conclusion, due to the noisy estimated distribution, naively combining NCD and long-tail methods cannot effectively solve the long-tailed novel class discovery problem.

## E   More results on Herbarium19 dataset

Fig.6 presents the distribution of the Herbarium19 dataset. The dataset is composed of 683 classes, out of which 178 categories have less than 20 samples, which presents a significant challenge when attempting to cluster novel classes. Tab.11 shows the average accuracy over both class and instance. Our proposed method outperforms the typical NCD methods by a considerable margin in both metrics, demonstrating the effectiveness of our approach.

## F   More explanation about estimation the number of novel categories

In order to better analyze the experiment of estimating the number of novel categories under a higher $K^u$, we visualize the novel class confusion matrices of UNO and Ours for known and unknown $K^u$ with $R^s = R^u = 50$ in ImageNet100. The y-axis is groud-truth, and the x-axis is prediction.

Table 11: Long-tailed novel class discovery performance on Herbarium19. We report average class and samples accuracy on test datasets. The top three lines is the accuracy average over classes. The bottom three lines is the accuracy average over samples.

| | Herbarium | | |
| Method | All | Novel | Known |
| --- | --- | --- | --- |
| Autonovel | 34.58 | 9.96 | 59.30 |
| UNO | 47.47 | 34.50 | 60.58 |
| Ours | **49.21** | **36.93** | **61.63** |
| Autonovel | 40.83 | 14.15 | 65.98 |
| UNO | 50.20 | 31.40 | 67.51 |
| Ours | **55.66** | **41.15** | **69.33** |

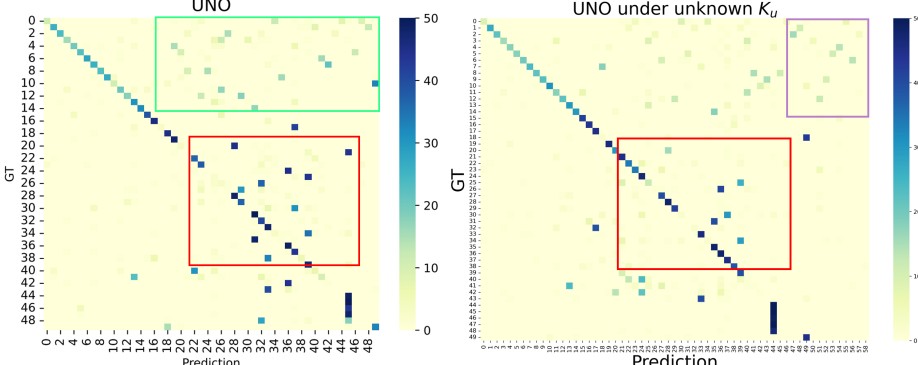

Figure 7: The confusion matrix of UNO for known and unknown $K^u$

UNO utilizes the Sinkhorn algorithm to generate pseudo labels, which enforces an equal distribution for each class. This will result in splitting a head class into multiple subcategories, which may be mixed up with the medium and tail classes, as illustrated by the green-bordered boxes in the left side image in Figure 7. Due to the misclassification of the head class, this will introduce noise that affects the quality of prediction for the medium and tail classes, as indicated by the red-bordered boxes in the left side image in Figure 7. When $K^u$ is larger than the true $K^u$, this problem can be alleviated because the head class is more likely to be assigned to categories that do not match the ground truth, as shown by the purple-bordered boxes in the right diagram. In this case, the noise in the pseudo labels for the medium class and tail class will be reduced, thus improving the accuracy of medium and tail classes, as shown in the the red-bordered boxes in the right side image in Figure 7.

According to the result analysis, we find that our method's novel classes accuracy will not be significantly affected when $K^u$ is greater than the true value. Our method learns a long-tailed distribution based on the training set data, dynamically adjusting the allocation ratio for novel classes. Compared to UNO, our method can increase the weight of the head class, which can suppress the decomposition of head classes into multiple subcategories. As a result, only a small number of categories are allocated to categories that do not match the ground truth, as shown the purple-bordered boxes in the right side image in Figure 8.

## G   Without strong model on CIFAR and ImageNet

We conduct experiment on CIFAR100 and ImageNet100 datasets using a non-strong model. We first perform unsupervised pre-training of the model on a long-tailed dataset using MoCoHe et al. (2020) on known classes. Next, we conduct supervised training on the known classes data. Finally, we discover the novel classes by jointly training on the known and novel classes.

The results in Tab.12 demonstrate that our method outperforms existing methods on the novel classes in both CIFAR100 and ImageNet100 datasets, especially in the more challenging setting where $R^u = 100$. However, on the CIFAR dataset, when equipped with the LA or cRT technique, the baseline method surpasses our

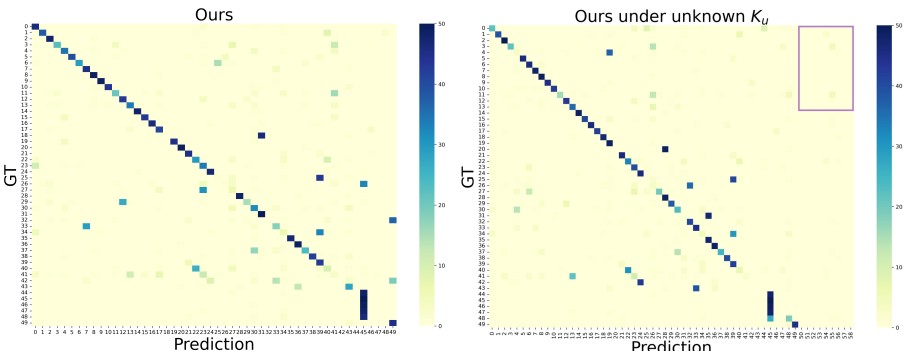

Figure 8: The confusion matrix of Ours for known and unknown $K^u$

Table 12: The performance on CIFAR-100, ImageNet100 without strong model.

| | **CIFAR100-50-50** | | | | | | **ImageNet100-50-50** | | | | | |
| | $R^s = 50, R^u = 50$ | | | $R^s = 50, R^u = 100$ | | | $R^s = 50, R^u = 50$ | | | $R^s = 50, R^u = 100$ | | |
| **Method** | **All** | **Novel** | **Known** | **All** | **Novel** | **Known** | **All** | **Novel** | **Known** | **All** | **Novel** | **Known** |
|---|---|---|---|---|---|---|---|---|---|---|---|---|
| AutoNovel | 29.72 | 16.90 | 42.54 | 30.39 | 16.82 | 43.96 | 45.52 | 25.24 | 65.80 | 42.88 | 19.56 | 66.20 |
| AutoNovel+LA | 30.74 | 18.60 | 42.88 | 30.54 | 17.04 | 44.04 | 45.90 | 25.28 | 66.52 | 43.04 | 19.88 | 66.20 |
| AutoNovel+cRT | 30.72 | 18.38 | 43.06 | 29.35 | 16.20 | 42.50 | 46.84 | 27.20 | 66.48 | 42.78 | 21.20 | 64.36 |
| UNO | 33.80 | 27.04 | 40.56 | 33.05 | 24.64 | 41.46 | 43.52 | 26.88 | 60.16 | 42.00 | 24.88 | 59.12 |
| UNO+LA | 34.78 | 27.50 | 42.06 | 34.66 | 24.04 | 45.28 | 45.78 | 29.08 | 62.48 | 44.80 | 26.80 | 62.80 |
| UNO+cRT | 36.98 | 29.44 | 44.52 | 33.27 | 25.50 | 41.04 | 43.72 | 27.36 | 60.08 | 43.16 | 25.92 | 60.40 |
| Ours | 36.42 | 30.22 | 42.62 | 35.08 | 27.36 | 42.80 | 47.66 | 29.48 | 65.84 | 47.08 | 27.96 | 66.20 |
| Ours+LA | 37.51 | 30.72 | 44.30 | 35.84 | 27.50 | 44.18 | 48.90 | 29.28 | 68.52 | 48.04 | 27.80 | 68.28 |

method on the known classes, resulting in our method being slightly better or worse than existing methods in overall accuracy. While on ImageNet100 datasets, our method surpasses the existing method by a sizeable margin. Furthermore, when applying the LA technique to our method, our results consistently outperform the existing methods in terms of the "All" and "Novel" metrics.

## H Known data are balanced

We conduct the experiment when known classes are balanced on CIFAR100 and ImageNet100. The results in Tab.13 show we achieve consistent and significant improvement on novel classes. For CIFAR100, our method achieves large improvements in different $R^u$ settings, surpasses the previous SOTA method by 5.74% and 6.20% in novel accuracy. For ImageNet100, we achieve a significant improvement over the previous SOTA method by 13.68% and 14.48% in novel accuracy.

Table 13: Experiments on balanced known classes

| | **CIFAR100-50-50** | | | | | | **ImageNet100-50-50** | | | | | |
| | $R^s = 1, R^u = 50$ | | | $R^s = 1, R^u = 100$ | | | $R^s = 1, R^u = 50$ | | | $R^s = 1, R^u = 100$ | | |
| **Method** | **All** | **Novel** | **Known** | **All** | **Novel** | **Known** | **All** | **Novel** | **Known** | **All** | **Novel** | **Known** |
|---|---|---|---|---|---|---|---|---|---|---|---|---|
| Autonovel | 55.66 | 24.24 | 87.08 | 54.60 | 22.24 | 86.96 | 69.20 | 45.28 | 93.12 | 66.60 | 39.60 | 93.60 |
| Autonovel + LA | 56.20 | 24.80 | 87.70 | 55.11 | 22.46 | 87.76 | 69.00 | 44.72 | 93.28 | 66.54 | 39.48 | 93.60 |
| AutoNovel + cRT | 57.11 | 27.78 | 86.44 | 56.01 | 25.88 | 86.41 | 69.71 | 45.74 | 93.68 | 67.35 | 41.23 | 93.48 |
| UNO | 59.84 | 32.88 | 86.80 | 57.19 | 27.16 | 87.22 | 68.68 | 44.04 | 93.32 | 67.84 | 41.60 | 94.08 |
| UNO + LA | 59.87 | 32.92 | 86.82 | 58.07 | 28.90 | 87.24 | 68.82 | 44.28 | 93.36 | 68.00 | 41.60 | 94.40 |
| UNO + cRT | 60.35 | 34.38 | 86.32 | 57.74 | 28.46 | 87.02 | 69.74 | 45.88 | 93.60 | 67.66 | 40.88 | 94.44 |
| Ours | 63.19 | 40.12 | 86.26 | 60.55 | 35.10 | 86.00 | 76.50 | 59.56 | 93.44 | 74.86 | 56.08 | 93.64 |

