# OpenReview forum: "Novel Class Discovery for Long-tailed Recognition"
_TMLR — Accepted by TMLR_

### Review · Reviewer_2okT · 2023-05-13

**Summary Of Contributions:**

For the novel class discovery, this paper presents a new setting that the distribution of novel class and known class is long-tailed. Then they propose a iterative algorithm consisting of equiangular prototype, adaptive self-labeling to alleviate the biases result from imbalance in known classes and novel classes.  The results on three datasets demonstrate the proposed method outperforms AutoNovel and UNO under such a new setting.


**Audience:**

Yes

**Broader Impact Concerns:**

There is no concern on the ethical implications of the work.

**Claims And Evidence:**

Yes

**Requested Changes:**

Some parts of the paper is not clear enough. Please see the weakness part. There also miss some detailed discussions of the experimental results.

**Strengths And Weaknesses:**

Strength:
1. This paper proposes a realistic and challenging setting for novel class discovery, where the distribution of novel and known classes is long-tailed.
2. The proposed method is effective and the details about the algorithm are provided.
3. The experiments are convincing and the ablation study analysis the effect of each component.

Weakness:
1. Compared with the results under known $K_u$, the proposed method and UNO perform better when the $K_u$ is unknown. Please give more explanation about the experiment of estimation the number of novel categories;
2. There are several confusing equations, such as \mathbf{Y}^u\mathbf{1}_{K_u} = \mathbf{\mu}  in equation (6), Y_u\in \mathbb{R}_+^{K_{u}\times M}, and \mathbf{1}_{K^u}\in \mathbb{R}_+^{K_{u}\times 1};
3. There are several typos, such as "We typically generate pseudo labels in a min-batch mode" in page 7.

---

> ### Comment · Editors_In_Chief · 2023-06-05
> **Response to Reviewer 2okT**
>
> **Q1: More explanation about the experiment of estimation the number of novel categories**
>
> To analysis the experiment of unknown $K^u$, we report the detailed results of UNO and Ours on ImageNet100-50-50 with known and unknown $K^u$ in the following table:
>
> |        |          | $R^s=50, R^u=50$ |      |       |       |        |       | $R^s = 50, R^u = 100$ |       |       |       |        |       |
> |--------|----------|--------------------|------|-------|-------|--------|-------|---------------------|-------|-------|-------|--------|-------|
> | Method |          | All                | Novel | Seen  | Head  | Medium | Tail  | All                 | Novel | Seen  | Head  | Medium | Tail  |
> | UNO    | $K^u$=50 | 65.30              | 43.08 | 87.52 | 44.93 | 49.50  | 32.13 | 62.52               | 37.84 | 87.20 | 49.73 | 43.10  | 18.93 |
> | UNO    | $K^u$=59 | 67.96              | 48.48 | 87.44 | 44.00 | 56.40  | 42.40 | 64.16               | 41.24 | 87.08 | 34.80 | 53.00  | 32.00 |
> | Ours   | $K^u$=50 | 73.94              | 61.48 | 86.40 | 77.47 | 55.80  | 53.07 | 69.38               | 51.96 | 86.80 | 77.47 | 54.20  | 23.47 |
> | Ours   | $K^u$=59 | 74.06              | 61.04 | 87.08 | 64.53 | 71.60  | 44.00 | 68.94               | 50.84 | 87.04 | 67.33 | 44.50  | 39.60 |
>
> Note that the numbers 50 and 59 correspond to the ground truth and estimated number of novel classes, respectively.
>
> UNO employs the Sinkhorn algorithm to generate pseudo labels, which enforces an equal distribution constraint for each class. However, due to the presence of a long-tailed data distribution, this equal-size constraint tends to divide the head class into multiple subcategories, resulting in a mixture of head, medium, and tail classes. Consequently, misclassification of the head class adversely affects the prediction quality for the medium and tail classes. Nonetheless, when the estimated value of $K^u$ exceeds the true value, this issue can be partially mitigated as the head class is more likely to be assigned to the additional categories. In such cases, the noise in the pseudo labels for the medium and tail classes is reduced, leading to improved accuracy for those classes.
>
> In contrast, our method learns a long-tailed distribution from the training dataset and dynamically adjusts the allocation ratio for novel classes. Compared to UNO, our approach assigns greater weight to the head class, thereby suppressing the decomposition of head classes into multiple subcategories. As a result, only a small number of categories are allocated to additional categories, thus achieving similar results as the setting of $K^u$ is ground truth.
>
> To facilitate a more comprehensive analysis of the experiment on estimating the number of novel categories, we visualize the novel class confusion matrices and provide a detailed explanation of UNO and our method for known and unknown $K^u$ in Appendix F.
>
> On the other hand, as indicated in the updated Table 5 in the main text, for CIFAR100 and Herbarium19 datasets where the estimated $K^u$ is lower than the ground truth, we observe our results are lower compared to when the true value of $K^u$ is used.
>
> **Q2: Confusing equations:**
>
> We appreciate the reviewer feedback. We have corrected the wrong matrix size of $\mathbf{Y}^u$ in Eq(6)(8) and highlighted the changes in red.
> In our method, $\mathbf{Y}^u$ means the pseudo labels of novel class, which is a matrix of size $M\times K^u$. $M$ refers to the number of samples stored in the buffer, and $K^u$ refers to the number of novel class. $\mathbf{1}_{K^u}$ refers to a all ones vector of size $K^u×1$.
>
> **Q3: Several typos:**
>
> We thank the reviewer's suggestion and have corrected the typo in the main text.

---

### Review · Reviewer_QARH · 2023-05-22

**Summary Of Contributions:**

The paper deals with an interesting and more realistic setup for novel class discovery (NCD), i.e. the case where discovery happens on an imbalanced dataset. It proposes a self labeling method that takes imbalance into account. They propose an algorithm for assigning pseudolabels to the unlabeled data by formulating it as an optimal transport problem.

**Audience:**

Yes

**Claims And Evidence:**

No

**Requested Changes:**

* For the ImageNet and CIFAR datasets, new experiments should be run where the pretrained model hasnt explicitly seen all images from the truncated classes, eg starting from a Places pretrained model.

* Experiments on all three datasets should be added for the cases where one further predicts the number of unknown classes, not just ImageNet100

* Experiments on iNaturalist would make this paper's claims much stronger. Right now the only real LT dataset used is Herbarium, where the are generally small (less than 2% overall) versus simply forgetting about class imbalance and using UNO.

* Would the proposed approach work if the known data were not LT? Known classes should not be expected to be LT as well - we have a large number of datasets that are balanced that would still be nice to use for LT discovery. It would be nice to have an experiment on the smaller datasets (ImageNet and CIFAR) where the known classes are balanced.


Other notes and questions:
- Missing refs on class discovery for long tail instance segmentation: Wang et al CVPR 2021: Unsupervised Discovery of the Long-Tail in Instance Segmentation Using Hierarchical Self-Supervision
- Were the hyperparameters tuned per dataset and per method? E.g. why is the UNO baseline worse after adapting it to LT for Herbarium? Were the hyperparameters properly tuned for that dataset?
- "Superivsed" type in Sec 2
- how much does using Eq9 offer? How much better is it than using a real-valued vector for w?

**Strengths And Weaknesses:**

Strengths:

* the paper deals with a more realistic extension for NCD - the assumptions and datasets used for the task make it highly unrealistic in practice and more of a toy task; this paper takes a step towards a more realistic task to solve

* The paper further partially studies the case of unknown number of unseen classes

* the proposed Adaptive Self-labeling Algorithm contains technical novelty, at least in the NCD setting.

Weaknesses:

* The proposed baseline for estimating the number of novel categories is not adequately described. If I understood correctly it requires rerunning the algorithm for many values of K_u and keep the one that performs best based on Eq11. Is that correct? What is the computational cost of that? how do you sample possible values for K_u to evaluate? It seems infeasible in practice to me, but maybe I am missing something.

* All experiments start from a very strong model, a DINO ViT trained on ImageNet1K, i.e. a dataset that contains a large number of classes including all "unseen" classes to discover for ImageNet100 and most of CIFAR100. Not only has the pretrained model seen those classes, which is generally fine cause it was trained without labels, however the model has however seen a much much higher number of images for all tail classes. So the premise now is that we first see many images per class, train for hundreds of epochs with a self-supervised loss and then retrain for a few epochs with LT labels. For the experiment to be fair, the pretrained model should not have seen the classes - a common protocol in LT papers is to start from a Places-pretrained model. This would be a much more fair setting I think

* The "new setup" presented by the authors boils down to using LT datasets for evaluation. I would expect a more extensive and thorough SoTA reproduction and comparison. The authors only compare two 2 typical NCD methods. They cite more recent ones, eg Yang et al., 2022a, but do not include in the comparison. Adding more methods and maybe also some baselines would make the paper stronger.

* The algorithm still is only evaluated on relatively small-sized dataset (CIFAR100, ImageNet100) that were artificially turned long-tail and a realistic, but still small/medium scaled dataset Herbarium19. No test on larger LT datasets like iNaturalist17/18 that have thousands of classes and are shown to be harder.

---

> ### Comment · Editors_In_Chief · 2023-06-05
> **Reponse to Reviewer QARH: Part 1**
>
> We appreciate the valuable feedback provided by the reviewer, and we addressed the main concern as follows.
>
> **Q1: The baseline for estimating the number of novel categories:**
>
> We appreciate the suggestions provided by the reviewer. The estimation of the number of novel categories is conducted separately from the main training process. To determine the optimal value for $K^u$, we adopt a baseline approach that combines hierarchical clustering and binary search. Initially, we set a relatively high value for $K^u$, such as 1000 for CIFAR100 and ImageNet100. We then perform a binary search to identify the value that maximizes the average accuracy. In each search step, we apply hierarchical clustering to the labeled and unlabeled data. This hierarchical clustering process is relatively efficient, and for datasets like CIFAR100 with 12.8k data points, the baseline approach takes approximately 35 minutes to estimate the optimal $K^u$ on the CPU. Furthermore, we can enhance the clustering algorithm's speed by leveraging the faiss package with GPU acceleration. To provide clearer explanations of the baseline approach, we have made revisions to Sections 4.5, Section 5.3, and Appendix A.
>
>
> **Q2: Conduct experiments without strong model on CIFAR and ImageNet:**
>
> We appreciate the reviewer's suggestion. We conduct experiment on CIFAR100 and ImageNet100 datasets without strong model. Following the typical training strategy in NCD (e.g., Autonovel and UNO), we first perform unsupervised pre-training of the model on the long-tailed known class dataset using MoCo [1]. Next, we conduct supervised training on the known classes data. Finally, we discover the novel classes by jointly training on the known and novel classes.
>
> The results in the below demonstrate that our method outperforms existing methods on the novel classes in both CIFAR100 and ImageNet100 datasets, especially in the more challenging setting where $R^u=100$. However, on the CIFAR100 dataset, when equipped with the LA or cRT technique, the baseline methods surpass our method on the known classes, resulting in our method being slightly better or worse than existing methods in overall accuracy. While on ImageNet100 datasets, our method surpasses the existing method by a sizeable margin.
>
> Furthermore, when applying the LA technique to our method, our results consistently outperform the existing methods in terms of the "All" and "Novel" metrics.
>
> |CIFAR100               | $R^s=50, R^u=50$ |           |       | $R^s=50, R^u=100$ |        |        |
> |---------------|------------------|-----------|-------|-------------------|--------|--------|
> |    Method           | All              | Novel     | Known | All               | Novel  | Known  |
> | AutoNovel     | 29.72            | 16.90     | 42.54 | 30.39             | 16.82  | 43.96  |
> | AutoNovel+LA  | 30.74            | 18.60     | 42.88 | 30.54             | 17.04  | 44.04  |
> | AutoNovel+cRT | 30.72            | 18.38     | 43.06 | 29.35             | 16.20  | 42.50  |
> | UNO           | 33.80            | 27.04     | 40.56 | 33.05             | 24.64  | 41.46  |
> | UNO+LA        | 34.78            | 27.50     | 42.06 | 34.66             | 24.04  | 45.28  |
> | UNO+cRT       | 36.98            | 29.44     | 44.52 | 33.27             | 25.50  | 41.04  |
> | Ours          | 36.42            | 30.22     | 42.62 | 35.08            | 27.36  | 42.80  |
> | Ours+LA       | 37.51            | 30.72     | 44.30 | 35.84             | 27.50  | 44.18  |
>
> | ImageNet | $R^s=50, R^u=50$ |        |       | $R^s=50, R^u=100$ |        |        |
> |---------------|----------------|--------|-------|-----------------|--------|--------|
> |               | All            | Novel  | Known | All             | Novel  | Known  |
> | AutoNovel     | 45.52          | 25.24  | 65.80 | 42.88           | 19.56  | 66.20  |
> | AutoNovel+LA  | 45.90          | 25.28  | 66.52 | 43.04           | 19.88  | 66.20  |
> | AutoNovel+cRT | 46.84          | 27.20  | 66.48 | 42.78           | 21.20  | 64.36  |
> | UNO           | 43.52          | 26.88  | 60.16 | 42.00           | 24.88  | 59.12  |
> | UNO+LA        | 45.78          | 29.08  | 62.48 | 44.80           | 26.80  | 62.80  |
> | UNO+cRT       | 43.72          | 27.36  | 60.08 | 43.16           | 25.92  | 60.40  |
> | Ours          | 47.66          | 29.48  | 65.84 | 47.08           | 27.96  | 66.20  |
> | Ours+LA       | 48.90 | 29.28 | 68.52 | 48.04 | 27.80 | 68.28 |

---

> > ### Comment · Editors_In_Chief · 2023-06-05
> > **Reponse to Reviewer QARH: Part 2**
> >
> > **Q3: More SoTA reproduction and Comparison:**
> >
> > The method developed by Yang et al. (2022a), referred to as ComEx, is based on the work of Fini et al. (2021), known as UNO. As shown in the table below, ComEx achieves a slight improvement over the results reported in the paper by Fini et al. (2021). However, it falls behind the results obtained from the official repository of UNO, which incorporates better augmentation techniques and longer training.
> > |Method|CIFAR100-20|CIFAR100-50|
> > |-|-|-|
> > |UNO(paper)|85.0 $\pm$ 0.6|52.9 $\pm$ 1.4|
> > |ComEx|85.7 $\pm$ 0.7|53.4 $\pm$ 1.3|
> > |UNO(repo)|90.4 $\pm$ 0.2|60.4 $\pm$ 1.4|
> >
> > (Note that CIFAR100-20 and CIFAR100-50 mean splitting 20 and 50 novel classes from CIFAR100, respectively. The results of UNO (paper) and ComEx are copied from Yang et al., 2022a, and UNO (repo) is obtained by running their official code.)
> >
> > Therefore, we chosen AutoNovel and UNO(repo) as our baseline methods for NCD, as they represent two typical approaches for clustering novel classes: pair-wise similarity and self-labeling. Additionally, we have incorporated typical long-tailed methods such as LA and cRT into our baseline framework. By combining these different approaches, we believe that our baseline is strong and our comparison is convincing.
> >
> > **Q4: Larger LT datasets like iNaturalist18:**
> >
> > We employ the DINO pretrained model and perform experiments on the larger iNaturalist18 datasets [2]. Due to the limited computational resources, we randomly sample 1000 and 2000 classes from the 8142 classes in iNaturalist18, which is large enough in the literature of novel class discovery [3,4]. Then, we split half of the sampled classes as novel classes, and append the details of the dataset to Table 1 in the main text. The results in the table below demonstrate that our method outperforms the baseline by 10% on novel classes, showcasing the superiority of our approach. Notably, the improvement observed on the iNaturalist18 dataset is more pronounced than on the Herbarium19 dataset. We have added the results to the Table 3 in main paper.
> >
> >
> > | Method | iNaturalist-1K|        |       |iNaturalist-2K | | |
> > |-----------------|-------------------|--------|--------|-------------------|--------|--------|
> > |                 | All               | Novel  | Known |  All               | Novel  | Known |
> > | Autonovel       | 42.33             | 11.67  | 73.00 | 39.08             | 8.57  | 69.60 |
> > | Autonovel + LA  | 42.40             | 11.27  | 73.53 | 44.67             | 14.33  | 75.00 |
> > | Autonovel + cRT | 44.20             | 16.13  | 72.27 | 37.95             | 9.27  | 66.63 |
> > | UNO             | 52.93             | 31.60  | 74.27 | 45.60             | 19.97  | 71.23 |
> > | UNO + LA          | 46.63           | 24.33 | 74.60 | 46.63             | 20.33  | 72.93 |
> > | UNO + cRT         | 51.73             | 32.60  | 70.87 | 46.47             | 24.90  | 68.03 |
> > | Ours            | 58.87             | 45.47  | 72.27 | 49.57             | 34.13  | 65.00 |

---

> > > ### Comment · Editors_In_Chief · 2023-06-05
> > > **Reponse to Reviewer QARH: Part 3**
> > >
> > > **Q5: The number of unknown classes for three datasets:**
> > >
> > > We conduct experiments on the CIFAR100 and Herbarium19 datasets using the estimated value for $K^u$. As shown below, our method demonstrates substantial improvements in novel class accuracy as well as overall class accuracy, except for the CIFAR dataset when $R^u=50$, where our method achieves comparable results.
> > > On the CIFAR dataset, the baseline surpasses our method on the known classes, particularly when combined with the LA or cRT technique, resulting in our method achieving slightly better or worse overall accuracy compared to existing methods in overall accuracy. When our method is combined with LA, we achieve better results. On the Herbarium19 dataset, our method outperforms existing methods by a significant margin, particularly in terms of novel class accuracy. We have included these results in Table 5 in the main text and adjusted the corresponding analysis accordingly.
> > >
> > > | CIFAR100-50-50  | $R^s$=50, $R^u$=50 |        |       | $R^s$=50, $R^u$=100 |        |        |
> > > |-----------------|-------------------|--------|-------|--------------------|--------|--------|
> > > |                 | All               | Novel  | Known | All                | Novel  | Known  |
> > > | Autonovel       | 41.25             | 16.08  | 66.42 | 43.74              | 17.64  | 69.84  |
> > > | Autonovel + LA  | 41.82             | 16.18  | 67.46 | 43.82              | 18.08  | 69.56  |
> > > | Autonovel + cRT | 45.81             | 19.50  | 72.12 | 43.44              | 18.18  | 68.70  |
> > > | UNO             | 47.67             | 28.92  | 66.42 | 46.56              | 25.92  | 67.20  |
> > > | UNO+LA          | 49.51             | 28.18  | 70.84 | 48.02              | 25.70  | 70.34  |
> > > | UNO+cRT         | 49.35             | 30.82  | 67.88 | 45.49              | 26.68  | 64.30  |
> > > | Ours            | 49.03             | 32.66  | 65.40 | 48.89              | 33.06  | 64.72  |
> > > | Ours + LA       | 49.64             | 32.46  | 66.82 | 49.87              | 33.12  | 66.62  |
> > >
> > > | Herbarium19  | |        |       |
> > > |-----------------|-------------------|--------|--------|
> > > |                 | All               | Novel  | Known |
> > > | Autonovel       | 37.84             | 15.64  | 60.16 |
> > > | Autonovel + LA  | 42.15             | 19.06  | 65.37 |
> > > | Autonovel + cRT | 40.95             | 18.86  | 63.16 |
> > > | UNO             | 40.83             | 23.55  | 58.24 |
> > > | UNO+LA          | 42.83             | 23.02  | 62.56 |
> > > | UNO+cRT         | 40.67             | 22.84  | 58.63 |
> > > | Ours            | 44.20             | 29.01  | 59.34 |
> > > | Ours + LA       | 45.74             | 29.55  | 61.90 |
> > >
> > > **Q6: Known data are not LT:**
> > >
> > > We conduct the experiment when known classes are balanced on CIFAR100 and ImageNet100. The results in below table show we achieve consistent and significant improvement on novel classes. We have appended the results to Appendix H.
> > >
> > > | CIFAR100-50-50  | $R^s$=1, $R^u$=50 |        |       | $R^s$=1, $R^u$=100 |        |        |
> > > |-----------------|-------------------|--------|-------|--------------------|--------|--------|
> > > |            | All           | Novel  | Known | All            | Novel  | Known  |
> > > | AutoNovel  | 55.66         | 24.24  | 87.08 | 54.60          | 22.24  | 86.96  |
> > > | Autonovel + LA | 56.20 | 24.80 | 87.70 | 55.11 | 22.46 | 87.76 |
> > > | Autonovel + cRT | 57.11 | 27.78 | 86.44 | 56.01 | 25.88 | 86.41 |
> > > | UNO        | 59.84         | 32.88  | 86.80 | 57.19          | 27.16  | 87.22  |
> > > | UNO+LA | 59.87 | 32.92 | 86.82 | 58.07 | 28.90 | 87.24 |
> > > | UNO+cRT | 60.35 | 34.38 | 86.32 | 57.74 | 28.46 | 87.02 |
> > > | Ours       | 63.19         | 40.12  | 86.26 | 60.55          | 35.10  | 86.00  |
> > >
> > >
> > > | ImgNet50-50     | $R^s$=1, $R^u$=50 |       |       | $R^s$=1, $R^u$=100 |       |       |
> > > | --------------- | ----------------- | ----- | ----- | ------------------ | ----- | ----- |
> > > |                 | All               | Novel | Known | All                | Novel | Known |
> > > | AutoNovel       | 69.20             | 45.28 | 93.12 | 66.60              | 39.60 | 93.60 |
> > > | Autonovel + LA  | 69.00             | 44.72 | 93.28 | 66.54              | 39.48 | 93.60 |
> > > | Autonovel + cRT | 69.71             | 45.74 | 93.68 | 67.35              | 41.23 | 93.48 |
> > > | UNO             | 68.68             | 44.04 | 93.32 | 67.84              | 41.60 | 94.08 |
> > > | UNO+LA          | 68.82             | 44.28 | 93.36 | 68.00              | 41.60 | 94.40 |
> > > | UNO+cRT         | 69.74             | 45.88 | 93.60 | 67.66              | 40.88 | 94.44 |
> > > | Ours            | 76.50             | 59.56 | 93.44 | 74.86              | 56.08 | 93.64 |
> > >
> > > **Q7: Missing ref**
> > >
> > > We thank the reviewer's suggestion and have add the missing ref to Sec 2.

---

> > > > ### Comment · Editors_In_Chief · 2023-06-05
> > > > **Reponse to Reviewer QARH: Part 4**
> > > >
> > > > **Q8: Hyperparameters tuning for per dataset and per method:**
> > > >
> > > > The hyperparameters for each method and dataset have been carefully tuned using a known-class validation set. When equipped with logits adjustment (LA), the accuracy of known classes is improved consistently. However, due to the noisy nature of the estimated distribution of novel classes, LA does not always effectively correct the bias in logits and can even have a detrimental effect, especially for challenging datasets like Herbarium19. In our analysis of the Herbarium19 dataset, we discover several empty sets in UNO's predictions, resulting in a relatively large value ($-\log(1e-8)$) being added to the empty classes after LA, which negatively impacts the results. If we have filtered out the empty sets, the updated results are as follows:
> > > > | Herbarium19  | All |   Novel     | Known |
> > > > |-----------------|-------------------|--------|--------|
> > > > | UNO             | 47.47             | 34.50  | 60.58 |
> > > > | UNO             | 37.22             | 12.15  | 62.40 |
> > > > | UNO+LA(filter)          | 46.76             | 27.96  | 65.69 |
> > > >
> > > > Upon analyzing the updated results, we have observed a noticeable improvement in the performance of known classes when using LA. However, there is still a significant drop in the performance of novel classes, which can be attributed to the noisy nature of the estimated distribution.
> > > >
> > > > **Q9: Supervised typo:**
> > > >
> > > > We thank the reviewer's suggestion and have corrected the typo.
> > > >
> > > > **Q10: ablation on $\mathbf{w}$:**
> > > >
> > > > As presented in Table 8 of the main text, we have conducted an ablation study to examine the impact of different designs of $\mathbf{w}$. The results demonstrate that the parametric $\mathbf{w}$, which is a function of $\tau$, outperforms a real-valued probability vector by a significant margin. The use of a real-valued probability vector tends to assign disproportionately large cluster sizes to certain clusters, which can have a negative impact on the overall performance.
> > > >
> > > >
> > > > **Reference:**
> > > >
> > > > [1] He, Kaiming and Fan, Haoqi and Wu, Yuxin and Xie, Saining and Girshick, Ross. Momentum contrast for unsupervised visual representation learning. pp 9729--9738, 2020.
> > > >
> > > > [2] Grant Van Horn, Oisin Mac Aodha, Yang Song, Yin Cui, Chen Sun, Alex Shepard, Hartwig Adam, Pietro Perona, and Serge Belongie. The inaturalist species classification and detection dataset. In Proceedings of the IEEE conference on computer vision and pattern recognition, pp. 8769–8778, 2018.
> > > >
> > > > [3] Kai Han, Sylvestre-Alvise Rebuffi, Sebastien Ehrhardt, Andrea Vedaldi, and Andrew Zisserman. Autonovel: Automatically discovering and learning novel visual categories. IEEE Transactions on Pattern Analysis and Machine Intelligence, 2021.
> > > >
> > > > [4] Vaze, Sagar and Han, Kai and Vedaldi, Andrea and Zisserman, Andrew. Generalized category discovery. Proceedings of the IEEE/CVF Conference on Computer Vision and Pattern Recognition, pp. 7492–7501, 2022.

---

### Review · Reviewer_fTg4 · 2023-05-25

**Summary Of Contributions:**

The paper has the following contributions:
1. A new setting for novel class discovery, where classes are long-tailed.
2. A new adaptive self-labeling method, which assigns pseudo labels to novel classes adaptively. The method casts the pseudo label generation problem to an optimal transport problem.
3. The authors conduct empirical experiments to demonstrate the effectiveness of the proposed method.

**Audience:**

Yes

**Claims And Evidence:**

Yes

**Requested Changes:**

Please follow the weaknesses part and polish the method section in a more rigorous way. Also, please make it more clear which techniques are existing, which techniques are novel, and better motivate the novel parts.

**Strengths And Weaknesses:**

Strengths:
1. Considering the novel class discovery under the long-tailed setting seems to be realistic and new.

Weakness:
1. The writing of the method section is unclear and confusing.
 - (5)(6) and (7)(8) describe two optimization problems. However, its not clear from the equations that (1) Are they minimization or maximization problems? Please use "min" or "max" instead of $\mathcal{L}$ as the main objective. (2) What are the variables to be optimized? Please mark those variables under "min" or "max".
- Why is the problem bi-level optimization? Can't the variables be jointly optimized?
- The use of "random variable" and "distribution" is very messy: For example, (1) in section 4.3, "Formally, we denote the cluster size distribution as $\nu \in \mathbb{R}_{+}^{K^u}$", I guess $\nu$ here is a random variable, not distribution. (2) The authors then "introduce an auxiliary variable $w$", and again mention $\nu$ is a prior distribution. Then how can the KL divergence be calculated between a random variable $w$ and a distribution $\nu$"? (3) Given the authors describe $w$ as an auxiliary variable $w$, the statement "then we normalize $w_i$ ... to make it a valid probability distribution" is contradictory. Is $w$ a random variable or distribution?
- In C. Mini-Batch Buffer, "min-batch" appears twice. Is it typo or something else?
- Section 4.5 is over-simplified with only textual description.

2. The novelty and motivation of the proposed components are weak:
- Section 4.1. The ETF seems to be a direct application of the original method proposed by Yang et al., 2022b
- Section 4.3. It seems the problem formulation largely follows Asano et al., 2020. The only difference is the additional auxiliary variable. Moreover, the parametric form of $w$ is very heuristic.

---

> ### Comment · Editors_In_Chief · 2023-06-05
> **Response to Reviewer fTg4: Part 1**
>
> We appreciate the valuable feedback provided by the reviewer, and we addressed the main concern as follows.
>
> ## The writing of the method section is unclear and confusing
> **Q1: The clarity of optimization problem:**
>
> (5)(6) and (7)(8) are both minimization problems, and we have revised the equation to make it clear.
>
> **Q2: Bi-level optimization:**
>
> Our adaptive self-labeling learning framework learns the pseudo labels and the representation in an alternating manner. And we formulate the process of generating pseudo labels as a relaxed optimal transport problem, as shown in Equ. (7) and (8). To solve the relaxed optimal transport problem, we introduce an additional variable, denoted as $\mathbf{w}$, and propose a bi-level optimization approach that alternates between optimizing $\mathbf{Y}$ and $\mathbf{w}$. Our bi-level optimization strategy is both simple and effective, as it leverages the efficient Sinkhorn Algorithm [1] for optimizing $\mathbf{Y}$ and employs simple gradient descent for optimizing $\mathbf{w}$. The additional variable allows us to incorporate prior knowledge of the long-tailed distribution, such as reparameterization.
>
> If we jointly optimize $\mathbf{Y}$ and $\mathbf{w}$, one possible way is to rewrite the formulation as:
>
> $$  \\min\\limits_{\\mathbf{Y}^u}\\quad \\langle \\mathbf{Y}^u,-\\mathbf{P}^{u\\top}\\mathbf{Z}\\rangle_F + \\gamma KL(\\mathbf{Y}^{u\\top}\\mathbf{1}_M, \\nu) \\quad\\quad \\text{s.t.}\\quad \\mathbf{Y}^u\\in\\{\\mathbf{Y}^u\\in \\mathbb{R}^{M\\times K^u}\_+ |\\mathbf{Y}^u \\mathbf{1}\_{K^u}=\\mu\\}$$
>
> This is a standard convex optimization problem that can be solved using conventional convex optimization methods. However, it does not take advantage of the problem's specific structure, leading to high computation complexity. Alternatively, one could transform the aforementioned problem into an unbalanced optimal transport (OT) problem [2,3], and solve it using unbalanced OT methods[3,4]. These methods require specific implementations, and the exact computational complexity in our specific problem setting is unclear. It's worth noting that these techniques are not the primary focus of our current manuscript, but they pose interesting research challenges for future exploration.
>
> **Q3: The use of "random variable" and "distribution" is very messy:**
>
> We have revised the description for clarity. Specifically, we denote the cluster size distribution as a probability vector $\nu \in \mathbb{R}^{K^u}_{+}$. The variable $\mathbf{w}$ is not a random variable but a variable that needs to be optimized, and it is also a probability vector.
>
> **Q4: "min-batch" typo:**
>
> We have corrected the typo to "mini-batch".
>
> **Q5: Section 4.5 is over-simplified:**
>
> We appreciate reviewer's suggestion and add the details of algorithm for estimating the number of novel classes to Appendix A.
>
> **Reference**
>
> [1] Marco Cuturi. Sinkhorn Distances: Lightspeed Computation of Optimal Transport. Advances in neural information processing systems, 26, 2013.
>
> [2] Frogner, Charlie and Zhang, Chiyuan and Mobahi, Hossein and Araya, Mauricio and Poggio, Tomaso A. Learning with a Wasserstein loss. Advances in neural information processing systems, 28, 2015.
>
> [3] Dvurechensky, Pavel and Gasnikov, Alexander and Kroshnin, Alexey. Computational optimal transport: Complexity by accelerated gradient descent is better than by Sinkhorn’s algorithm. International conference on machine learning, pp 1367-1376. PMLR, 2018.
>
> [4] Yiling Luo, Yiling Xie, and Xiaoming Huo. Improved rate of first order algorithms for entropic optimal transport. In International Conference on Artificial Intelligence and Statistics, pp. 2723–2750. PMLR, 2023.

---

> > ### Comment · Editors_In_Chief · 2023-06-05
> > **Response to Reviewer fTg4: Part 2**
> >
> > ## The novelty and motivation of the proposed components are weak:
> >
> > **Q6: The difference with ETF method proposed by Yang et al.2022b:**
> >
> > The classifier tends to be biased towards majority classes in imbalanced scenarios. To address the issue of imbalance learning in supervised scenarios, Yang et al. (2022b) introduce the ETF classifier, which is motivated by the phenomenon of neural collapse. The problem of classifier bias becomes even more severe in the clustering of imbalanced novel data, where both the representation and classifier are learned without clean label information. Due to the absence of ground-truth for novel classes, how to apply the ETF classifier to handle novel class discovery is unclear. To this end, we leverage our adaptive self-labeling algorithm, and extend the ETF classifier to handle both known and novel classes, mitigating the imbalance learning of known and novel classes in a unified manner. We have revised Section 2 and Section 4.1 to provide further clarification on this point.
> >
> > **Q7: The difference with self-labeling method proposed by Asano et al.2020:**
> >
> > Asano et al. (2020) formulate the self-labeling problem as an optimal transport problem and utilize the Sinkhorn algorithm to solve it. However, as stated in section 4.3 (at the top of page 6), they assume an equal cluster size, which is unrealistic in many practical scenarios where the class distributions are imbalanced. Consequently, the equal-size assumption largely limits its application. To mitigate this issue, we relax the equal-size constraint and formulate the imbalance self-labeling problem as a relaxed optimal transport problem. Furthermore, as illustrated in Q2, we propose an efficient bi-level algorithm to solve the relaxed optimal transport problem.
> >
> > Our parameterization of $\textbf{w}$ incorporates prior knowledge of long-tailed distributions, and we empirically find that it fits well with real-world long-tailed datasets, such as Herbarium19 and iNaturalist18 datasets. Furthermore, as demonstrated in Table 8 (main text), our parameterization simplifies the optimization process, mitigating the risk of converging to local minima.
> >
> > Hence, the primary novelty of our approach stems from the relaxed optimal transport formulation for imbalanced self-labeling and the novel bi-level optimization algorithm developed to efficiently solve our formulated problem. In addition, we extend the ETF classifier to the novel class discovery.

---

### Decision · Action_Editors · 2023-07-26

**Recommendation:** Accept with minor revision

**Comment:**

This paper explores novel class discovery in a long-tailed distribution scenario, which is more realistic than previous relevant research, proposing an iterative self-labeling algorithm to address the class-imbalance issue. The extensive experimental results support the proposed algorithm, and the authors provided informative responses to the reviewers, leading to acceptance by two reviewers. One reviewer expressed a concern about the novelty but will feel OK if the paper is accepted. The AE weighed the pros and cons and recommends acceptance with minor revision. The authors are mandated to make vital revision to include the technical clarifications as well as the supplemented experimental results in their responses.

**Audience:**

Yes

**Claims And Evidence:**

Yes